# CD8+ cells and small viral reservoirs facilitate post-ART control of SIV replication in M3+ Mauritian cynomolgus macaques initiated on ART two weeks post-infection

Olivia E. Harwood[1], Lea M. Matschke[2], Ryan V. Moriarty[1], Alexis J. Balgeman[1], Abigail J. Weaver[1], Amy L. Ellis-Connell[1], Andrea M. Weiler[3], Lee C. Winchester[4], Courtney V. Fletcher[4], Thomas C. Friedrich[2,3], Brandon F. Keele[5], David H. O'Connor[1,3], Jessica D. Lang[1,6], Matthew R. Reynolds[2,3], Shelby L. O'Connor[1,3]*

1 Department of Pathology and Laboratory Medicine, University of Wisconsin-Madison, Madison, Wisconsin, United States of America, 2 Department of Pathobiological Sciences, University of Wisconsin-Madison, Madison, Wisconsin, United States of America, 3 Wisconsin National Primate Research Center, Madison, Wisconsin, United States of America, 4 College of Pharmacy, University of Nebraska Medical Center, Omaha, Nebraska, United States of America, 5 AIDS and Cancer Virus Program, Frederick National Laboratory for Cancer Research, Frederick, Maryland, United States of America, 6 Center for Human Genomics and Precision Medicine, University of Wisconsin-Madison, Madison, Wisconsin, United States of America

* slfeinberg@wisc.edu

**Data Availability Statement:** RNA sequencing data were deposited in the NCBI's Gene Expression

## Abstract

Sustainable HIV remission after antiretroviral therapy (ART) withdrawal, or post-treatment control (PTC), remains a top priority for HIV treatment. We observed surprising PTC in an MHC-haplomatched cohort of MHC-M3+ SIVmac239+ Mauritian cynomolgus macaques (MCMs) initiated on ART at two weeks post-infection (wpi). None of the MCMs possessed MHC haplotypes previously associated with SIV control. For six months after ART withdrawal, we observed undetectable or transient viremia in seven of the eight MCMs, despite detecting replication competent SIV using quantitative viral outgrowth assays. *In vivo* depletion of CD8α+ cells induced rebound in all animals, indicating the observed PTC was mediated, at least in part, by CD8α+ cells. With intact proviral DNA assays, we found that MCMs had significantly smaller viral reservoirs two wpi than a cohort of identically infected rhesus macaques, a population that rarely develops PTC. We found a similarly small viral reservoir among six additional SIV+ MCMs in which ART was initiated at eight wpi, some of whom exhibited viral rebound. These results suggest that an unusually small viral reservoir is a hallmark among SIV+ MCMs. By evaluating immunological differences between MCMs that did and did not rebound, we identified that PTC was associated with a reduced frequency of CD4+ and CD8+ lymphocyte subsets expressing exhaustion markers. Together, these results suggest a combination of small reservoirs and immune-mediated virus suppression contribute to PTC in MCMs. Further, defining the immunologic mechanisms that engender PTC in this model may identify therapeutic targets for inducing durable HIV remission in humans.

Omnibus and are publicly available under GEO accession GSE225770.

**Funding:** The Wisconsin National Primate Research Center is supported by grants P51RR000167 and P51OD011106. National Institutes of Health NIH R01 AI108415 supported OEH, AJB, AJW, ALE-C, and SLO. CVF and LCW were supported by R01 AI124965. RVM was supported by the National Institute of Allergy and Infectious Diseases of the National Institutes of Health under Award Number T32AI55397. BFK was supported with federal funds from the National Cancer Institute, National Institutes of Health, under Contract No. 75N91019D00024/ HHSN261201500003I. The content of this publication does not necessarily reflect the views or policies of the Department of Health and Human Services, nor does mention of trade names, commercial products, or organizations imply endorsement by the U.S. Government. The funders had no role in study design, data collection and analysis, decision to publish, or preparation of the manuscript.

**Competing interests:** The authors have declared that no competing interests exist.

## Author summary

The ultimate goal of human immunodeficiency virus (HIV) therapeutic development is to enable HIV+ individuals to stop taking daily lifelong antiretroviral therapeutics (ART) and remain in virological remission, an outcome called post-treatment control. Post-treatment control is challenging to study because it is very rare in humans. Thus, there are currently no therapeutics that lead to post-treatment control. Simian immunodeficiency virus (SIV) in rhesus macaques is the most widely used preclinical model for HIV. However, post-treatment control is also uncommon in rhesus macaques. The mechanisms that govern post-treatment control remain largely unknown because PTC is so rare. Our lab observed surprising post-treatment control in a cohort of Mauritian cynomolgus macaques that were initiated on ART two weeks post-infection. While Mauritian cynomolgus macaques have been used for SIV research, they are not routinely used to study post-treatment control. We found that small viral reservoirs and immune-mediated virus suppression contribute to post-treatment control in Mauritian cynomolgus macaques. Moving forward, this model can be used to further define the mechanisms that engender post-treatment control as well as identify therapeutic targets in humans for inducing durable HIV remission.

## Introduction

Identifying the immunological and virological attributes that lead to antiretroviral therapy (ART)-free HIV remission is critical for establishing a functional cure. A rare group of people infected with HIV can suppress virus replication to ≤400 copies/mL for ≥16 weeks to several years after ART interruption. These individuals are termed post-treatment controllers (PTCs) [1–4]. While specific major histocompatibility complex (MHC) alleles are associated with spontaneous control of HIV without ART (elite control; EC), MHC alleles are not thought to influence the likelihood of becoming a PTC [2,5–9]. The underlying mechanisms that lead to PTC are poorly understood. Unfortunately, it is challenging to study PTC in humans since people with HIV rarely undergo treatment interruption, PTCs are uncommon, and PTC cohorts are heterogeneous in genetic composition, timing of ART initiation, and definition of PTC [10].

Among PTCs, the most common intervention is starting ART early after infection (reviewed in [11]). This phenomenon is best characterized in the VISCONTI, CHAMP, SPARTAC, and CASCADE cohorts, where 4–16% of participants became PTCs [1–4]. Unlike post-exposure prophylaxis, which can prevent establishment of systemic HIV infection if ART is initiated within 72 hours of exposure [12], initiating ART within weeks of exposure limits the size of cell-associated HIV-1 DNA reservoirs [13,14] and prevents the accumulation of viral variants, while preserving antiviral immune function [15,16]. However, not everyone who begins ART immediately after infection becomes a PTC; it is unclear why early ART potentiates PTC in some individuals and not others.

Because CD8+ T cells mediate viral load decline after peak viremia both on and off ART [17,18] and suppress viral replication during ART [19], CD8+ T cells may also contribute to PTC. Unlike the well-established role of CD8+ T cell-mediated virus suppression observed in most ECs [20,21], PTCs generally have low numbers of IFN-γ-producing HIV-specific CD8 + T cells that poorly suppress HIV replication ex vivo [2]. While T cell exhaustion can predict time to rebound and precede loss of viral control [22,23], the association between T cell subsets

expressing exhaustion markers and PTC has not been thoroughly evaluated in existing PTC studies. Alternatively, both cytolytic (e.g., directly killing target cells) and non-cytolytic (e.g., transcriptional silencing) CD8+ T cell activity preserved by ART may be essential for PTC [24]. Due to limitations in evaluating HIV-specific CD8+ T cells, it has not been possible to fully define the contributions of CD8+ T cells to PTC in humans. Therefore, a reproducible animal model of PTC would enable experimental interventions like CD8 depletion to be performed, and antigen-specific CD8+ T cells could be evaluated because animals can share MHC alleles or haplotypes.

Rhesus macaques (RMs) infected with simian immunodeficiency virus (SIV) are the most common nonhuman primate model of HIV [25], yet few become PTCs [26,27]. Definitions of PTC vary between studies but commonly include maintaining viremia at $<10^4$ copies/mL for months to years post ART withdrawal. Initiating RMs on ART within the first five days of infection has been shown to lead to productive infection and result in viral rebound in 100% (9/9) of the RMs within three weeks of ART interruption [13,28]. However, initiating RMs on ART within the first five days of infection can also resemble post-exposure prophylaxis, resulting in RMs that do not rebound after ART interruption but also do not appear to be productively infected [26,29]. Initiating ART within 6–12 days after infection results in productive infection but rarely leads to PTC; Okoye and colleagues found that 32/33 RMs exhibited viral rebound, most within 2–6 weeks after ART withdrawal with plasma viremia $>10^4$ copies/mL [26]. Since the SIV RM model of HIV has failed to produce enough PTCs to unravel the underlying mechanisms of PTC, a different model may be required to establish and evaluate PTC.

In the present study, we describe frequent PTC in Mauritian cynomolgus macaques (MCMs). MCMs are an exotic macaque population with limited genetic diversity due to a small number of invasive founder animals colonizing Mauritius within the last several hundred years. The limited genetic diversity extends to the MHC, with only seven MHC haplotypes (M1-M7) [30] in the entire population. While the M1 and M6 MHC haplotypes are associated with spontaneous SIV control without ART, the M3 MHC haplotype is not [8,30,31]. We initiated ART at two weeks post-infection (wpi) in eight M3+ MCMs that did not have the M1 or M6 haplotypes and discontinued treatment eight months later. For six months after ART withdrawal, plasma viremia in seven of the eight MCMs was either undetectable or only transiently detectable at $<10^4$ copies/mL. We find that MCMs had smaller acute reservoirs than similarly infected RMs two wpi. Experimental depletion of CD8α+ cells in all seven PTC MCMs led to viral rebound, demonstrating a role for these cells in maintaining ART-free viral control. M3+ MCMs can now be used to dissect the mechanisms by which small viral reservoirs and CD8α+ cells facilitate PTC.

## Results

### Prolonged control of SIV replication after ART withdrawal in M3+ MCMs initiated on ART at two wpi

We previously infected eight MCMs intravenously (i.v.) with 10,000 infectious units (IUs) of SIVmac239M and started them on antiretroviral therapy (ART) two weeks later (Fig 1A; MCM_2wk cohort). None of these MCMs possessed the M1 or M6 MHC haplotypes associated with SIV control (Table 1) [8,30,31]. We included animals possessing at least one copy of the M3 MHC haplotype because this haplotype is not associated with spontaneous SIV control. We anticipated that animals possessing at least one copy of the M3 haplotype and no copies of the M1 or M6 haplotypes would not naturally suppress SIV replication. Four animals (denoted by open symbols) previously received therapeutic interventions while four animals received only ART. These interventions included three heterologous viral vectors encoding SIV Gag

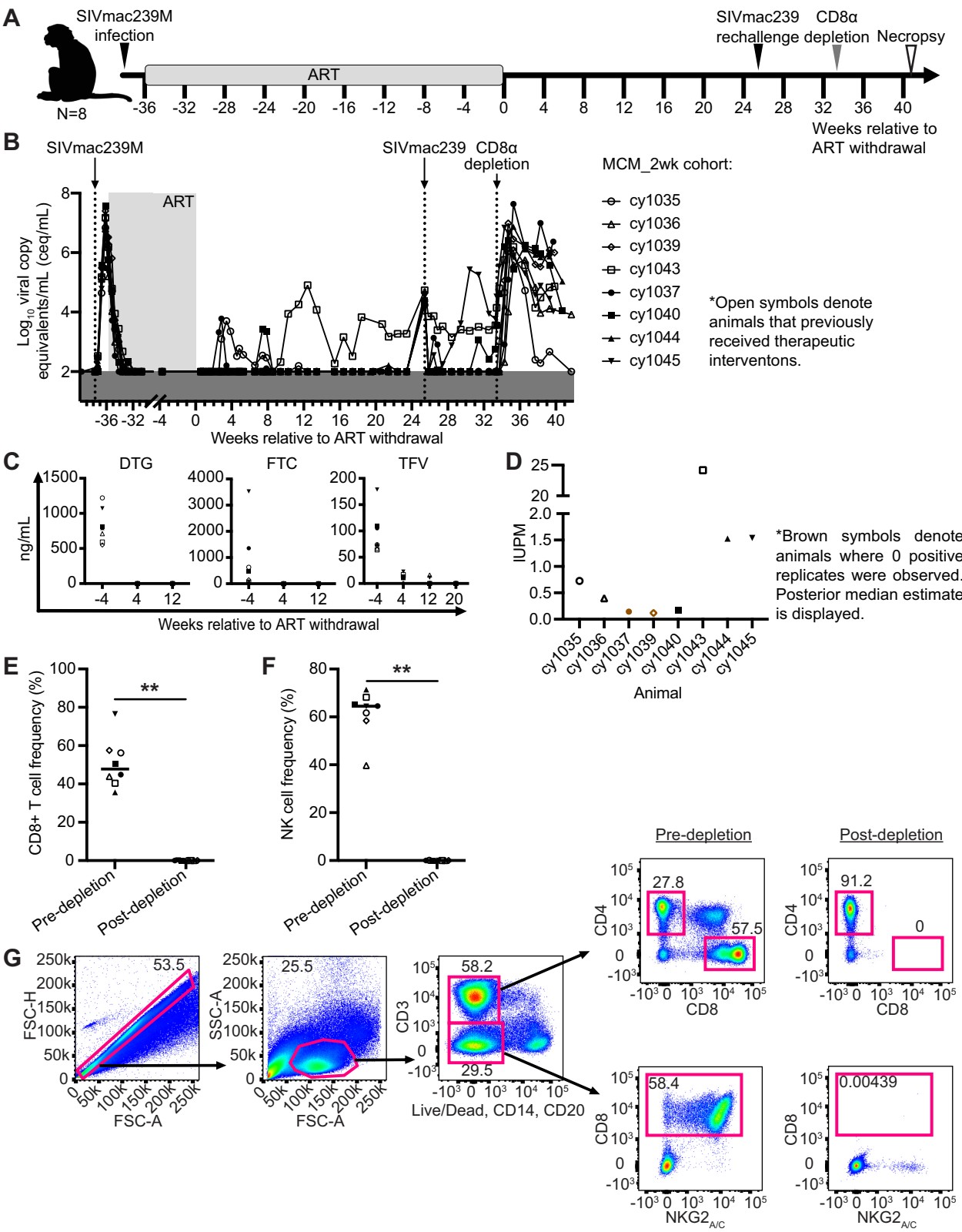

**Fig 1. Experimental design, ART drug concentrations, and immunological and virological analyses.** (A) Study design depicting the timeline of SIVmac239M infection, ART initiation, ART withdrawal, SIVmac239 rechallenge, administration of the CD8a depleting antibody, and necropsy.

Animals that previously received therapeutic interventions [32] are displayed as open symbols. (B) Individual plasma viral loads for the entire study. Viral loads are displayed as $\log_{10}$ceq/mL. (C) Plasma concentrations of DTG (left), FTC (center), and TFV (right) during ART treatment (the -4 week time point) and after ART withdrawal (4 and 12 week time points). Amounts BLQ are displayed as zero ng/mL. (D) IUPM for each animal ~19–21 weeks after ART withdrawal determined by QVOA. Square symbols denote animals where zero positive replicates were observed; posterior median estimate is displayed. (E-F) The frequency of CD8+ T cells (E) and NK cells (F) before and 10 days after administration of the CD8α depleting antibody. Results are displayed for each animal individually with the lines at the median. ** $P = 0.0078$. P values were calculated using Wilcoxon signed-rank tests. (G) Representative gating strategy used to evaluate CD8+ T cells (top images) and NK cells (bottom images) before (left) and 10 days after CD8α depletion (right). Animal shape graphics were created with BioRender.com.

delivered in four-week intervals beginning approximately 20 weeks post SIV and ending six weeks before ART withdrawal. The vaccinated animals then received three doses of the IL-15 superagonist N-803 separated by two weeks each, beginning 3 days after ART withdrawal. The immunological effects of the therapeutic interventions are described elsewhere [32], but did not appear to impact the results described herein. All eight animals received ART for approximately eight months before ART was interrupted. Up to 23 weeks after ART withdrawal, only one animal (cy1043, 12.5%) displayed sustained viral rebound that began nine weeks after ART withdrawal (Fig 1B). The other seven animals (87.5%) maintained undetectable or transiently detectable viremia at $<10^4$ copies/mL.

To investigate whether PTC was due to the residual presence of ART drugs, we measured plasma concentrations of DTG, TFV, and FTC. All three drugs were detectable in the normal range seen in humans [33,34] during ART and dropped below the limit of quantification (BLQ) by one to five months post-ART withdrawal (Fig 1C).

Next, we performed quantitative viral outgrowth assays (QVOAs) with PBMCs collected ~19–21 weeks after ART withdrawal to determine if replication-competent SIV remained. SIV *gag* was detected by qRT-PCR in at least one replicate in six of the eight animals (Fig 1D), indicating that reactivatable SIV was present.

## Rechallenge with an isogenic virus is largely contained in PTCs

Approximately 25 weeks after ART withdrawal, all eight MCMs were rechallenged i.v. with SIVmac239 to determine whether they possessed immune responses that could protect from rechallenge. SIV *gag* RNA was detected in plasma collected within 70 minutes of administering SIVmac239, demonstrating that SIV was infused. Four animals (cy1037, cy1040, cy1043, and cy1045) either rebounded or had transient detectable blips in viremia within six weeks of rechallenge while viremia in the other four animals (cy1035, cy1036, cy1039, and cy1044)

**Table 1. Animals in MCM_2wk cohort.**

| Animal ID | Sex | Species | MHC | ART initiation time |
|---|---|---|---|---|
| cy1035 | M | Cynomolgus | *Mafa-A1*063* M3/M4 | 2 weeks |
| cy1036 | M | Cynomolgus | *Mafa-A1*063* M3/M5 | 2 weeks |
| cy1037 | M | Cynomolgus | *Mafa-A1*063* M3/recM3M4[a] | 2 weeks |
| cy1039 | M | Cynomolgus | *Mafa-A1*063* M3/M3 | 2 weeks |
| cy1040 | M | Cynomolgus | *Mafa-A1*063* M3/M3 | 2 weeks |
| cy1043 | M | Cynomolgus | *Mafa-A1*063* M2/recM3M4[a] | 2 weeks |
| cy1044 | M | Cynomolgus | *Mafa-A1*063* M2/M3 | 2 weeks |
| cy1045 | M | Cynomolgus | *Mafa-A1*063* M3/M5 | 2 weeks |

[a]Expressed the MHC class I A and B and MHC class II DRB and DQ alleles present in the M3 MHC haplotype, but also expressed MHC class II DP alleles of the M4 MHC haplotype.

remained undetectable (Fig 1B). IFN-γ ELISPOT assays indicate that after rechallenge, the animals with undetectable viremia had more positive T cell responses (measured by the number of spot-forming cells [SFCs] per $1 \times 10^6$ PBMCs) to a pool of Gag peptides and two known Mafa-A1*063-restricted T cell epitopes ($Gag_{386-394}GW9$ and $Nef_{103-111}RM9$) than animals with detectable post-rechallenge viremia (S2 Fig). These results suggest that post-rechallenge viral control may be associated with higher SIV-specific cellular immune responses.

## CD8α+ immune cells mediate PTC

Eight weeks after SIVmac239 rechallenge, we transiently depleted CD8α+ cells *in vivo* from all eight MCM_2wk animals (Fig 1A) by intravenously administering the MT807R1 CD8α-depleting antibody. We observed a ~3–6 $log_{10}$ increase in SIV viremia in all animals within ten days of administration of the CD8α-depleting antibody (Fig 1B), corresponding with a significant decline in the frequency of CD8+ T cells and NK cells (Fig 1E–1G). These results suggest that post-ART immune control was mediated by CD8α+ cells.

## MCMs have smaller total and intact viral reservoirs than RMs at two wpi

It is widely believed that smaller viral reservoirs increase the likelihood of PTC [35–38]. Much of the integrated vDNA in HIV+ patients is defective [39] but assays that can quantify the replication-competent reservoir may better predict the likelihood of viral rebound after ART withdrawal [40]. Intact proviral DNA assays (IPDAs) estimate the number of cells containing vDNA and distinguish defective proviruses from intact proviruses, which are likely replication-competent [40]. We used IPDAs to compare the number of intact and total vDNA copies per $1 \times 10^6$ cell equivalents between MCMs at the time of ART initiation, and a separate cohort of RMs infected with SIVmac239M by the same dose and route [41]. RMs initiated on ART ~one to four weeks post SIV exhibit prompt viral rebound after ART interruption [18,26,35,42,43], but it is unknown if differences in reservoir sizes between RMs and MCMs account for the different rebound kinetics after ART withdrawal.

We found that RMs had significantly more total and intact vDNA than MCMs at two wpi (Fig 2A and 2B, respectively). The distribution of vDNA species within the total vDNA compartment was similar between RMs and MCMs (Fig 2C). Even though the RM viral loads at two wpi were approximately 1 $log_{10}$ higher than MCMs (Fig 2D and 2E), both species had similar peak viral loads (Fig 2D and 2F). RMs exhibited a more sustained peak while MCMs exhibited a rapid post-peak decline in viremia (Fig 2D), as evidenced by a larger area under the curve (AUC) from pre-SIV to two wpi (Fig 2G), and there are approximately 10-fold more cells containing intact proviral DNA in RMs compared to MCMs at two wpi. Unfortunately, the RMs never received ART and none of the following virological or immunological parameters could be compared to the MCM cohort after this time point.

To explore why MCMs had fewer vDNA copies per $1 \times 10^6$ cell equivalents than RMs, we evaluated memory phenotypes and CCR5 expression on CD4+ T cells between MCMs and RMs before SIV infection (Fig 2H–2L). In line with published observations [44], we found that MCMs had lower frequencies of CD4+ T cells (Fig 2H) that were biased toward effector and transitional memory phenotypes (Fig 2I). Despite having similar frequencies of CCR5+ CD4+ T cells (Fig 2J), MCMs exhibited a lower abundance of CCR5 receptors (measured by mean fluorescence intensity, MFI) on the surface of CD4+ T cells than RMs (Fig 2K and 2L). The lower CCR5 expression on CD4+ T cells of MCMs may contribute to reduced susceptibility to infection and may help explain the differential reservoir sizes between RMs and MCMs.

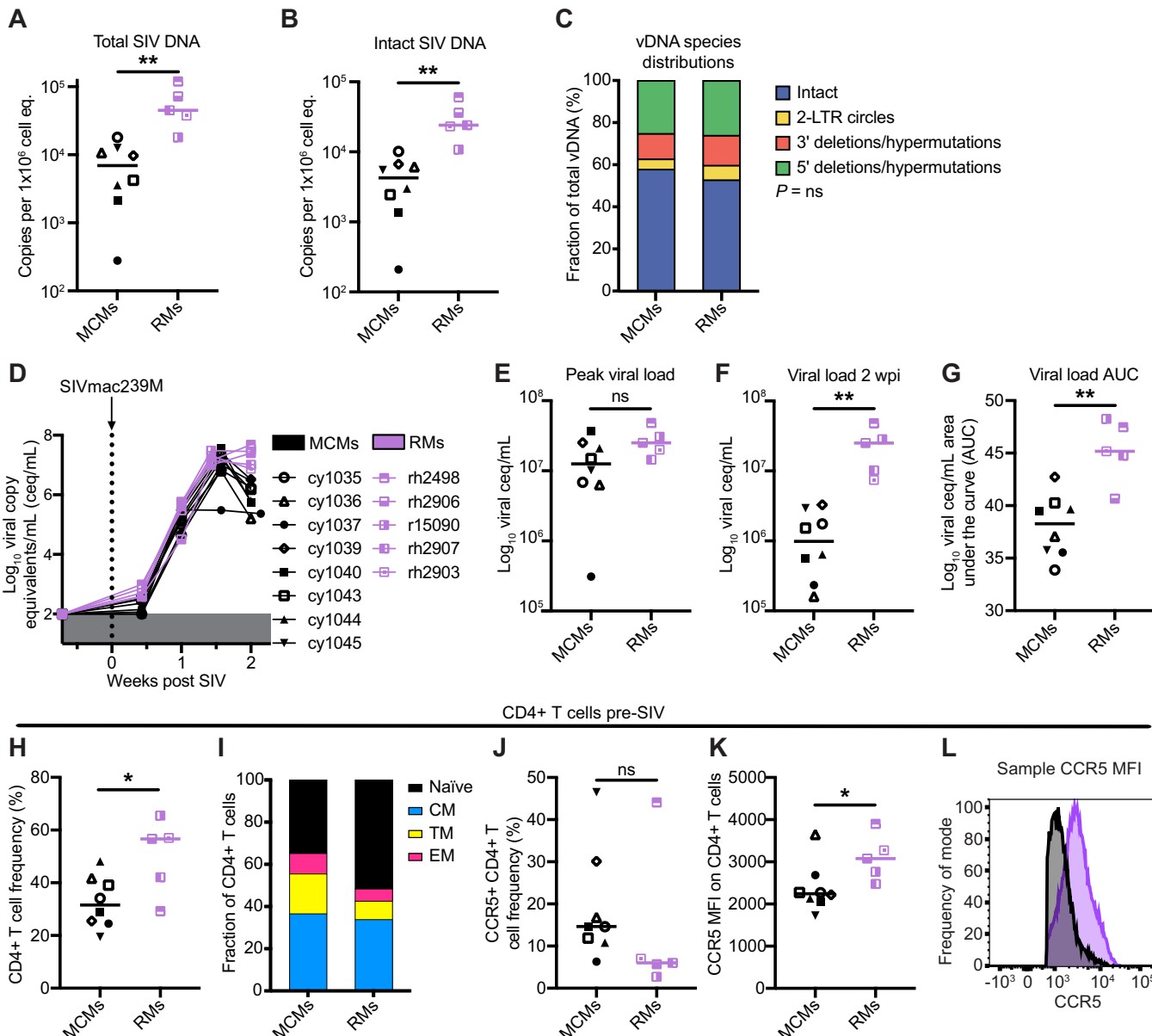

**Fig 2. Total and intact proviral DNA in the peripheral blood, CCR5 expression on peripheral blood CD4+ T cells, and viremia dynamics differ between MCMs and RMs during acute SIV infection.** (A-B) Total (A) and intact (B) vDNA per million cell equivalents in PBMCs from the eight MCMs (black) or five RMs (purple) collected two wpi. (C) The distribution of vDNA types. (D-G) Individual plasma viral loads from pre-SIV through two wpi (D), peak viral load (E), viral load two wpi (F), and viral load area under the curve (AUC) from SIV naïve through two wpi (G) for the MCMs and RMs. (H-L) Peripheral blood frequencies of CD4+ T cells (A), distribution of memory CD4+ T cell subsets (B), frequencies of CCR5+ CD4+ T cells (J), CCR5 mean fluorescence intensity (MFI) on bulk CD4+ T cells (K), and a sample histogram depicting CCR5 staining MFI (L) on one MCM (black) compared to one RM (purple). ns P≥0.05, ** P≤0.01. Mann-Whitney U tests were performed for A, B, E, F, G, H, J, and K and a Chi-square test was performed in C. Viral loads in D, E, and F are displayed as log$_{10}$ceq/mL. Median and individual values are shown in A, B, E, F, G, H, J, and K, and mean fractions are shown in C and I.

### A longer duration of SIV infection prior to ART does not appear to increase the frequency of cells with intact proviral DNA but increases the likelihood of viral rebound

We wanted to determine if MCMs were predisposed to having fewer cells with intact proviral DNA or if the frequency of these cells increased with a longer duration of infection prior to

ART initiation. We performed IPDAs on samples from a second cohort of six MCMs that were intrarectally infected with 3,000 TCID50 of SIVmac239 and started receiving ART at eight wpi (Fig 3A; MCM_8wk cohort). Unlike the MCM_2wk cohort, a subset of these animals expressed the M1 MHC haplotype associated with SIV control (Table 2) [8,30]. We were surprised that the number of intact proviral DNA copies per million cell equivalents was slightly lower in the MCM_8wk cohort at 8wpi (Median = $3.1x10^3$ copies per $1x10^6$ cell equivalents) compared to the MCM_2wk cohort at 2wpi (Median = $4.3x10^3$ copies per $1x10^6$ cell equivalents). Unfortunately, PBMCs collected at 2wpi from the MCM_8wk cohort were unavailable, so we could not make a longitudinal comparison within this cohort. Additionally, we were unable to statistically compare the amount of vDNA in the MCM_2wk cohort to the MCM_8wk cohort at the time of ART initiation due to the confounding variable that these groups were initiated on ART at two and eight wpi, respectively.

The size of the viral reservoir typically decays during ART, even though it is not eliminated [14,45]. We wanted to determine if the number of intact proviral DNA copies per million cell equivalents declined in MCMs during ART. We performed IPDAs on PBMCs collected during chronic ART, but prior to ART withdrawal (Fig 3B). In both MCM cohorts, there was a significant decline in total and intact vDNA (Fig 3C, 3D, 3F, and 3G). Statistical analyses were not performed on the vDNA species distributions because the on-ART distributions represent a small number of sequences near the limit of detection (Fig 3E and 3H).

## Despite small reservoir sizes, viral rebound can occur in MCMs initiated on ART at eight wpi

We hypothesized that the MCM_8wk cohort would also exhibit PTC because they had fewer cells containing intact proviral DNA than the MCM_2wk cohort at the time that ART was initiated. We discontinued ART for these six animals after ~20 months of ART and viral loads were monitored for 12 weeks. Four of the six MCM_8wk cohort animals (cy0721, cy0724, cy0726, cy0735, 66.7%) exhibited sustained post-ART viremia ($>5x10^2$ copies/mL) within four weeks of ART withdrawal and the other two animals (cy0719, cy0723, 33.3%) exhibited transiently detectable blips in viremia 6–12 weeks after ART withdrawal (Fig 4A). This contrasted the MCM_2wk cohort, where only one of eight animals developed sustained plasma viremia beginning nine weeks post ART withdrawal, three exhibited transient viral blips, and the remaining four maintained undetectable viremia (Fig 4B).

## Combining MCM_2wk and MCM_8wk cohorts and re-grouping by post-ART viremia kinetics

We re-categorized all 14 MCMs during the 12 weeks after ART interruption (eight from the MCM_2wk cohort and six from the MCM_8wk cohort) as post-ART controllers (maintained undetectable viremia), blippers (had transient viremia above the limit of detection), or rebounders (exhibited sustained viremia above $5x10^2$ copies/mL without re-control) (Fig 4C). We re-grouped the MCMs to evaluate whether post-ART viremia or PTC was associated with differences in CD8+ T cell phenotype or function. Combined plasma viral loads are displayed in Fig 4D. We categorized animals into the three groups using post-ART plasma viremia, though there were no differences in the cell-associated viral RNA (vRNA) in total lymph node (LN) homogenates across all animals (Fig 4E). There were also no differences in the plasma anti-SIVmac239 gp120 IgG antibodies between the groups during ART or after ART withdrawal (S3 Fig).

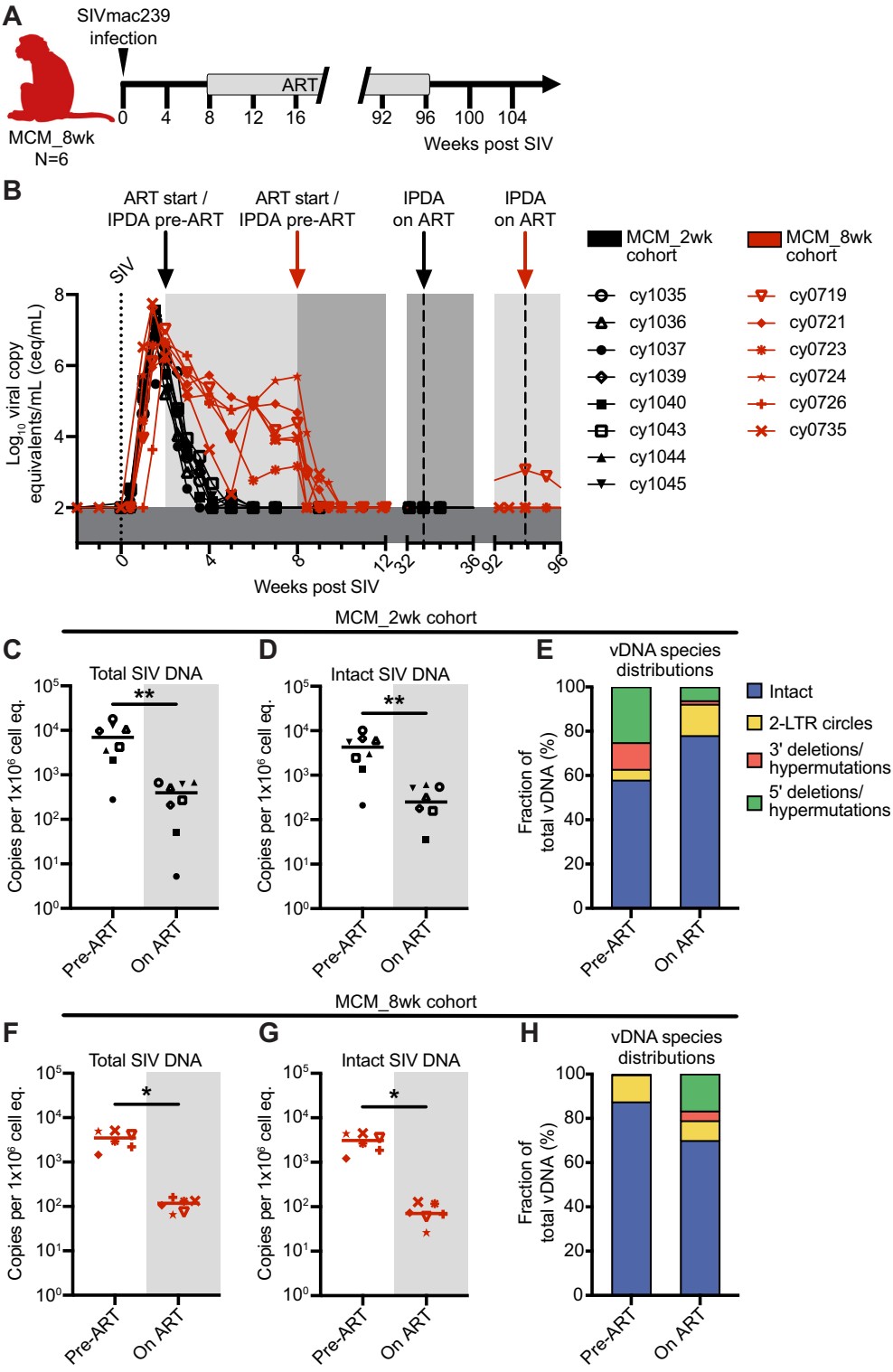

**Fig 3. Experimental design for the MCM_8wk cohort and total and intact proviral DNA in the peripheral blood of the MCM_8wk and MCM_2wk cohorts.** (A) Study design depicting the timeline of SIVmac239 infection, ART initiation, and ART withdrawal for the six MCM_8wk animals. (B) Individual plasma viral loads depicting the timing of ART initiation and time points for IPDA vDNA analyses for all 14 MCMs. (C-E) Total vDNA (C), intact vDNA (D), and the distribution of vDNA types (E) in the eight MCM_2wk animals before (2wpi, pre-ART) and 31 weeks after

ART initiation (33wpi, on ART). (F-H) Total vDNA (F), intact vDNA (G), and the distribution of vDNA types (H) in the six MCM_8wk animals before (eight wpi, pre-ART) and 86 weeks after ART initiation (94 wpi, on ART). * $P = 0.0312$, ** $P = 0.0078$. Wilcoxon signed rank tests were performed for C, D, F, and G. Median and individual values are shown in C, D, F, and G, and mean fractions are shown in E and H. Animal shape graphics were created with BioRender.com.

## PTC in MCMs is associated with preserved CD4+ frequencies and lower frequencies of CD8+ T cells expressing exhaustion markers

We hypothesized that high CD8+ T cell frequencies may have contributed to PTC, but we found no differences in the bulk CD8+ T cell counts or frequencies in PBMCs from the PTCs, blippers, or rebounders during ART or 12 weeks post ART withdrawal (Fig 4F and 4I). CD4 + T cell counts, frequencies, and the CD4:CD8 ratio significantly decreased in the rebounders after ART withdrawal (Fig 4G, 4H, and 4J). This was not surprising because increased viral replication is associated with CD4+ T cell decline in viral rebounders [46]. CD4+ T cell frequencies before ART withdrawal were similar between groups and did not predict PTC. To test the hypothesis that the PTCs had T cells with greater functional capabilities than the blippers or rebounders, we measured the frequency of CD4+ and CD8+ T cells expressing CD107a, TNF-α, and IFN-γ after Gag peptide pool stimulation *in vitro* (S4 Fig). Surprisingly, we found no differences in the frequencies of CD4+ or CD8+ T cells expressing combinations of one, two, or three cytokines/cytolytic markers during ART or 12 weeks post ART withdrawal (S4C and S4D Fig). We also measured differential gene expression in CD8+ T cells from four PTCs and four rebounders during ART (time point 1) and after ART withdrawal (time point 2; 12 weeks post ART withdrawal in the PTCs or the last time point with undetectable viremia prior to rebound in the rebounders). While there were marked changes in gene expression between time points, these changes were similar among PTCs and rebounders (S5 Fig). While there were no differences in gene expression between the controllers and rebounders at both time points evaluated, this could be due to the small number of animals.

Because T cell exhaustion has been implicated in HIV/SIV disease progression and loss of viral control (reviewed in [47]), we evaluated T cell subsets expressing exhaustion markers (S6–S8 Figs). The frequency of bulk CD8+ T cells, $T_{CM}$, $T_{TM}$, and $T_{EM}$ expressing CTLA4 significantly declined when comparing samples collected during ART treatment to 12 weeks post ART withdrawal in the blippers and PTCs (S6 Fig). The frequency of Gag-specific (Gag$_{386-394}$GW9 tetramer+) CD8+ T cells expressing LAG3 and CTLA4 also significantly declined in the PTCs after ART withdrawal (S7B and S7C Fig). Lastly, we observed a reduced frequency of CD4+ $T_{CM}$ and $T_{EM}$, and CD8+ $T_{CM}$ expressing PD1 in the PTCs after ART withdrawal (S8 Fig).

**Table 2. Animals in MCM_8wk cohort.**

| Animal ID | Sex | Species | MHC | ART initiation time |
|---|---|---|---|---|
| cy0719 | M | Cynomolgus | *Mafa-A1*063* M1, M3 | 8 weeks |
| cy0721 | M | Cynomolgus | *Mafa-A1*063* recM1/M3, M5[a] | 8 weeks |
| cy0723 | M | Cynomolgus | *Mafa-A1*063* M4, recM3/M1[b] | 8 weeks |
| cy0724 | M | Cynomolgus | *Mafa-A1*063* M1, M4 | 8 weeks |
| cy0726 | M | Cynomolgus | *Mafa-A1*063* M2, M3 | 8 weeks |
| cy0735 | M | Cynomolgus | *Mafa-A1*063* M1, M4 | 8 weeks |

[a]Expressed the MHC class I A allele present in the M1 MHC haplotype, but also expressed the MHC class I B and MHC class II DR, DQ, and DP alleles of the M3 MHC haplotype.

[b]Expressed the MHC class I A and B alleles present in the M3 MHC haplotype, but also expressed MHC class II DR, DQ, and DP alleles of the M1 MHC haplotype.

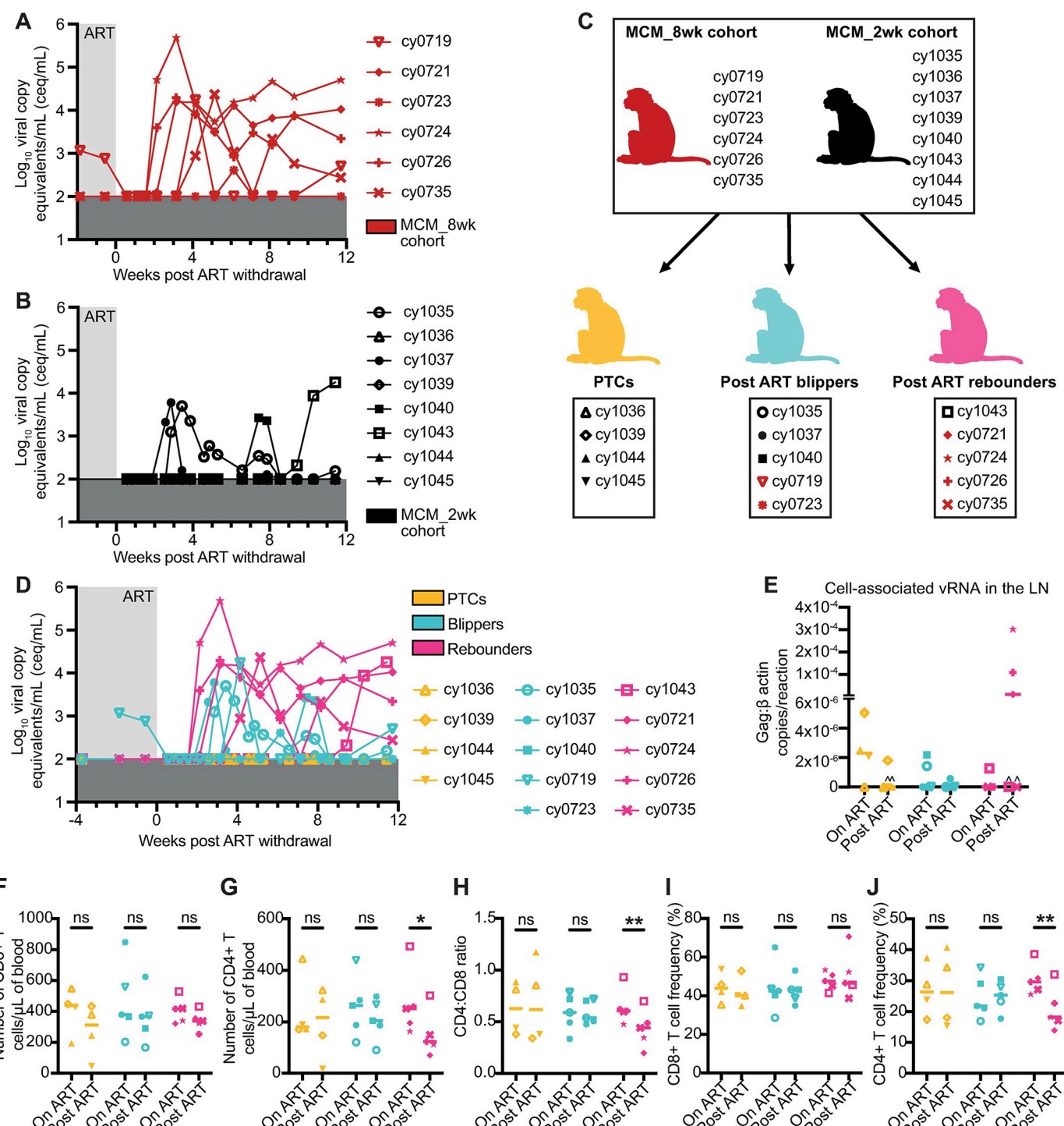

**Fig 4. Rearrangement of both animal cohorts into post-ART SIV PTCs, blippers, and rebounders, and differences in CD4+ and CD8+ T cell subsets.** (A-B) Individual plasma viral loads for the MCM_8wk cohort (A) and the MCM_2wk cohort (B) from two weeks before ART withdrawal to 12 weeks post ART withdrawal. Viral loads are displayed as $\log_{10}$ceq/mL. (C) Regrouping of animal cohorts into animals that, after ART withdrawal, became PTCs (orange), blipped (teal), or fully rebounded (pink). (D) Combined post-ART viral loads for the PTCs, blippers, and rebounders. (E) Cell-associated vRNA in the LNs of the animals during ART treatment (on ART) and 3 weeks post ART withdrawal (post ART). Results are displayed for each animal as the ratio of Gag:β actin per reaction with the lines at the median. Carrots denote samples where Gag was detectable but not quantifiable. (F) Number of CD8+ T cells per μL of blood, (G) number of CD4+ T cells per μL of blood, (H) CD4:CD8 ratio, (I) CD4+ T cell frequencies, and (J) CD8+ T cell frequencies in the PTCs (orange), blippers (teal), and rebounders (pink) during ART treatment (on ART) and 12 weeks post ART withdrawal (post ART). Results are displayed for each animal individually with the lines at the median. * $P \leq 0.05$, ** $P \leq 0.01$. P values were calculated using two-way ANOVA tests with Šídák correction for multiple comparisons. Animal shape graphics were created with BioRender.com.

## Discussion

Here, we provide the first example of consistent PTC of SIV in a cohort of M3+ MCMs infected with SIVmac239M and initiated on ART at two wpi. Although ART drugs had been cleared systemically (Fig 1C) and replication-competent virus was present (Fig 1D), seven of the eight M3+ MCMs had undetectable plasma viremia during the six months after ART withdrawal. Our results implicate both CD8α+ cells and the formation of unusually small viral reservoirs in post-ART SIV suppression.

Prompt viral rebound after CD8 depletion provides evidence that CD8+ cells are vital to maintaining PTC, which includes both CD8+ T cells and NK cells. A similar study using the CD8β depleting antibody could be used in the future to investigate the contributions of CD8 + T cells versus NK cells to PTC. Here, we investigated which CD8+ T cell parameters are associated with PTC. For example, elevated expression of immune checkpoint inhibitors (ICIs) like PD1, CTLA4, and LAG3 on T cells coincides with progressive loss of effector function, and this dysfunction has been implicated in the loss of viral control and disease progression (reviewed in [47]). We therefore measured the frequency of various T cell subsets expressing ICIs before and after ART withdrawal. Because the frequencies of some bulk, memory, and SIV-specific T cell subsets expressing certain ICIs declined after ART withdrawal compared to before in PTCs but not in animals that rebounded (S6–S8 Figs), we suggest that reduced expression of ICIs by T cells may help facilitate PTC.

Although we detected no significant differences in the frequency of SIV-specific cytokine-expressing CD8+ T cells in any animal group, there was a modest trend toward an increasing frequency of polyfunctional CD8+ T cells in the PTCs after ART withdrawal compared to before (S4 Fig). The frequencies of cytokine-producing cells were very low overall, confounding our ability to draw strong conclusions. However, if this trend were to remain consistent among additional animals, it is possible that polyfunctional cytokine-producing CD8+ T cells contribute to PTC in MCMs. Additional animal cohorts will be required to test this hypothesis.

LN resident CD8+ T cells may also contribute to PTC, specifically in containing viral replication in lymphoid viral reservoir sites. Previous studies have shown that EC is associated with lymphoid CD8+ T cell function and that HIV-specific lymphoid CD8+ T cell dysfunction coincides with aborted viral control [48,49]. Despite disparate post-ART viral load profiles, we observed similar amounts of cell-associated vRNA in LN homogenates (Fig 4E), suggesting the presence of an effective immune response suppressing viral replication in the peripheral LNs of the PTCs, blippers, and rebounders. It is possible that functional LN resident CD8+ T cells contributed to this virus suppression and that rebound viremia in the rebounders originated from other anatomical lymphoid sites (e.g., mesenteric LNs) or tissues (e.g., gut). Unfortunately, LN and gut tissues from all MCMs were not available for analyses of post-ART LN and tissue resident CD8+ T cell phenotype and function.

CD8+ T cells may also suppress virus replication through non-cytolytic mechanisms, such as suppressing virus transcription [24]. Our observation that three MCMs maintained undetectable plasma viremia between ART interruption and CD8 depletion indicates that either the CD8+ T cells were effectively killing infected CD4+ T cells without evidence of virus replication or SIV transcription was blunted. CD8+ T cells have been recently shown to promote HIV latency by boosting CD4+ T cell survival, quiescence, and stemness, primarily through increased TGF-β and WNT signaling within the infected CD4+ T cell [50]. Future studies should explore whether this type of CD8+ T cell virus-suppressive activity can contribute to PTC in this MCM model.

SIV Env-specific antibodies have recently been shown to correlate directly with time to rebound in a separate MCM study [51]. However, we did not observe any association between

bulk anti-SIVmac239 gp120 IgG antibodies between animals that did and did not become PTCs (S3 Fig). Substantial differences in animal study design (e.g., ART initiation time, time receiving ART, animal MHC) and Env antigens (gp140 versus gp120) between Wu and colleagues' study and our study may impact the formation and measurement of anti-SIV Env antibodies and therefore explain the discrepancy. Nonetheless, these discordant results warrant further investigation into the potential contribution of humoral immune responses to PTC in MCMs.

While neutralizing antibodies against HIV have been associated with viral control, macaques rarely generate potent neutralizing antibodies against SIVmac239 [52]. Neutralizing antibodies against SIVmac239 have been identified [53,54], but are uncommon. We cannot discount the possibility that neutralizing antibodies were involved in virus suppression, but we only evaluated env-binding antibodies in the present study. It would be interesting to examine anti-SIVmac239 neutralizing antibodies as a potential correlate of PTC in future studies of MCMs [55,56].

PTC may be influenced by events of acute infection. The earlier ART is initiated during acute infection, the higher the likelihood of later immunologic control after ART withdrawal [2,4]. This is partially attributed to limited reservoir establishment. Individuals with smaller viral reservoirs are overrepresented among identified PTCs, but the contribution of reservoir size to predicting PTC and time to rebound is unclear [2,38,57]. We found the number of cells containing total and intact vDNA was approximately 10-fold lower in MCMs than RMs in PBMCs collected at two wpi (Fig 2), suggesting that MCMs naturally develop smaller viral reservoirs than RMs. This could be a consequence of low CCR5 expression on MCM CD4+ T cells compared to RM CD4+ T cells (Fig 2) [58]. Within MCMs, we could not distinguish PTCs from rebounders by the number of cells containing intact or total vDNA at the time of ART initiation or after ART initiation (Figs 3 and 4). The rebounders may have had more vDNA in cells within tissues, like the gut, that comprise the majority of the viral reservoir, but these sites were not sampled. Additionally, type-I interferon production during acute infection can both positively and negatively impact SIV reservoir formation by enhancing innate control of SIV through modulating the expression of antiviral genes but increasing the availability of susceptible target cells for infection [59]. However, matched acute samples were unavailable for analysis of interferon production in the animals described here. These possibilities should be examined in future MCM studies. Collectively, MCMs consistently develop small viral reservoirs in PBMCs, but this is not sufficient to predict PTC.

One potential reason for the disparate vDNA reservoir sizes between MCMs and RMs could be SIVmac239's superior adaptation to RMs than MCMs. Previous studies have shown that host control mechanisms can limit the spread of cross-species transmitted viruses, even if they replicate effectively during acute infection [60,61]. SIVmac239, originally isolated from RMs, exhibits strong adaptation to this species. While MCMs can still develop peak and set-point viremia similar to that of RMs during untreated SIVmac239 infection [30,62], the related SIVmac251 strain is slightly less pathogenic in MCMs than RMs [63]. Further comparisons of SIVmac239 pathogenesis in MCMs and RMs during acute and chronic infection will offer valuable insights into the unique attributes of these two populations that may impact HIV/SIV cure studies.

PTC in humans is rare (typically estimated at 4–16% of HIV+ individuals [1,2,4]), and ~80% exhibited rebound within five years [1]. A relevant animal model would enable the immunological and virological correlates of PTC to be characterized. Consistent PTC in MCMs without known protective MHC alleles in this study aligns with the observation that HIV+ PTCs are not enriched for protective MHC alleles [2,8,9,30]. It is alternatively possible that the PTC observed is specifically associated with the M3 haplotype, even though this

haplotype is not associated with EC. This possibility should be addressed in additional studies of MCMs comprising additional genetic backgrounds.

In addition, Sharaf and colleagues recently found that HIV+ PTCs had nearly ten-fold less intact proviral DNA genomes than non-controllers just prior to ART interruption by IPDA [57]. Bender and colleagues similarly found that HIV+ humans on long term ART had approximately ten-fold less cells containing intact vDNA genomes than SIVmac251+ or SIVmac239 + RMs on long term ART [64]. While it is difficult to compare IPDA results between different species infected with different viruses, the viral reservoir size of SIV+ MCMs was approximately 10-fold less than SIV+ RMs (Fig 2), resembling the relationship between the viral reservoir sizes of HIV+ humans and SIV+ RMs. Our observations indicate the potential value of MCMs as a model of PTC.

Most RM ART interruption studies identify zero PTCs [18,26,35,42,43]. While the definition of PTC varies dramatically between studies [10], many definitions of PTC (e.g., maintain plasma viremia $<10^4$ copies/mL [27]) are lenient enough to include at least nine of the 14 MCMs described here. Our results indicate that PTC is much more common in M3+ MCMs than RMs. Notably, MCM_8wk cohort animals expressing the M1 haplotype associated with EC were also spread evenly between the blipper and rebounder groups, suggesting that while M1+ animals are predisposed to EC, the expression of the M1 haplotype did not appear to impact post-ART viral rebound kinetics here. Even the four MCM_8wk animals that rebounded within four weeks of ART interruption had peak rebound viremia of only 4.2–5.7 $\log_{10}$ceq/mL (Fig 4A). In contrast, RMs initiated on ART in similar timeframes consistently rebound within two weeks of ART interruption with peak rebound viremia of ~4–7 $\log_{10}$ceq/mL [18,26,35,42]. Thus, even when MCMs exhibit viral rebound, viremia is delayed and of a smaller magnitude compared to RMs. Although MCMs can develop high peak and set point viremia akin to that of RMs during untreated SIV infection [30], MCMs and RMs have distinct post-ART rebound kinetics. We propose that other differences in viral load kinetics and pathogenesis between these species may, therefore, also exist.

It would be interesting to directly compare MCMs to RMs in an ART interruption study. While acute CD8+ T cells do not impact reservoir establishment in RMs [18], it is possible that CD8+ T cells impair reservoir establishment in MCMs. Specific depletion of CD8+ T cells during acute SIV infection in MCMs could help determine if CD8+ T cells are required for reservoir formation, virus suppression under ART, and later PTC. Even though similar depletion studies were performed in RMs [18], acute depletion studies in this novel MCM model of PTC may identify key features of immune-mediated virus suppression that were otherwise confounded by the larger viral reservoir in RMs. Further, understanding why RMs generate more cells with total or intact vDNA during acute infection than MCMs may reveal key features of MCMs that limit infection. For example, CD4+ T cells in RMs may be more susceptible to SIV infection than CD4+ T cells in MCMs, which would affect reservoir formation and size. Therefore, future studies directly comparing RMs and MCMs would be important to better understand early reservoir formation, PTC, and which macaque models are best for studying a functional HIV cure.

The limited genetic diversity of MCMs permits a more precise evaluation of the genetic basis of CD8+ T cell responses associated with virus suppression than is possible in the more genetically diverse RMs. For example, RMs have 25 MHC-E alleles while humans have just three and MCMs have two [65,66]. CD8+ T cells recognize pathogen-derived peptides in the context of MHC-E and can protect against SIV [67]. Antigen-specific MHC-E-restricted CD8 + T cells can perform antiviral functions like secreting antiviral cytokines and killing infected cells [68], so these responses may contribute to PTC in MCMs but not RMs. This should be

evaluated in ART interruption studies of MCMs with different MHC-E backgrounds to identify the contributions of MHC-E alleles to CD8+ T cell responses in the context of PTC.

It would also be beneficial to attempt to "break" PTC in MCMs to characterize the contribution of CD8+ T cell function to PTC. Initiating ART at two wpi in MCMs may be the "goldilocks" time to allow antiviral immunity to develop but not become dysregulated. Therefore, if ART was initiated later in chronic infection (i.e., six months post SIV), then viral evolution, reservoir expansion, and immune exhaustion may prevent PTC. Alternatively, MCMs could be infected with an SIV strain containing mutations in CD8+ T cell epitopes [69] to determine if acute SIV-specific CD8+ T cells are required for subsequent PTC. An epitope knock-out SIV strain could exploit the limited MHC genetic diversity of MCMs to understand if acute antigen-specific CD8+ T cells contribute to PTC. Experiments aimed at preventing PTC in MCMs may help explain why MCMs become PTCs.

One drawback of this study was that some of the MCM_2wk animals received therapeutic interventions described elsewhere [32]. Immunological changes did occur in the vaccinated animals after each intervention, but these effects were transient, and largely limited to Gag-specific T cell responses in the peripheral blood. While it is possible that this vaccine regimen impacted the CD8+ T cell characteristics we measured in the present study (e.g., phenotype, function, gene expression), the interventions did not impact post-ART viremia kinetics. Because seven of the eight MCMs (three vaccinated and four unvaccinated) did not exhibit sustained post-ART viremia, it is unlikely that the therapeutic regimen impacted the outcome of PTC.

The MCM_8wk cohort (Fig 3) was also an imperfect match to the MCM_2wk cohort (Fig 1) due to differences in MHC genetics, route of SIV challenge, timing of ART initiation, and duration of ART suppression. It is possible that these disparities impacted the immune responses generated. However, all MCMs had similar amounts of total and intact vDNA in the PBMCs before ART initiation and similar amounts of cell-associated vRNA in the LNs, indicating that these animals were an appropriate comparator group (Figs 3D and 4E). Despite these limitations, this cohort was valuable to demonstrate that MCMs, as a specific population, do not always become PTCs.

In sum, we have identified a novel model of PTC using M3+ MHC haplomatched MCMs. Of the eight haplomatched animals in this study, seven became PTCs. Comparisons of this MCM_2wk cohort to a cohort of RMs and to a separate MCM cohort (MCM_8wk) revealed the importance of small viral reservoirs and phenotypically functional CD8+ T cell subsets for PTC. Interruption of ART in SIV+ MCMs leads to very different rebound kinetics (i.e., lower post-ART peak and set point viral load and longer time to rebound) compared to RMs described in the literature with similar infection histories [26]. The study design described here, consisting of SIV+ MCMs initiated on ART at two wpi and withdrawn from ART eight months later needs to be repeated in larger cohorts to further unravel the mechanisms of post-ART virus suppression. Future post-ART SIV remission studies using this novel MCM model of PTC may inform the development of therapeutic interventions.

## Materials and methods

### Ethics statement

All macaques were cared for by the staff at the Wisconsin National Primate Research Center (WNPRC) in accordance with the regulations, guidelines, and recommendations outlined in the Animal Welfare Act, the Guide for the Care and Use of Laboratory Animals, and the Weatherall Report. The University of Wisconsin-Madison College of Letters and Science and Vice Chancellor for Research and Graduate Education Centers Institutional Animal Care and

Use Committee (IACUC) approved the nonhuman primate research covered under IACUC protocols G005443 and G005507. The University of Wisconsin-Madison Institutional Biosafety Committee approved this work under protocols B00000117 and B00000205. All macaques were housed in standard stainless-steel primate enclosures providing required floor space and fed using a nutritional plan based on recommendations published by the National Research Council. Housing rooms were maintained at 65–75˚F, 30–70% humidity, and on a 12:12 light-dark cycle (ON: 0600, OFF: 1800). Animals were fed twice daily a fixed formula, extruded dry diet with adequate carbohydrate, energy, fat, fiber, mineral, protein, and vitamin content (Harlan Teklad #2050, 20% protein Primate Diet, Madison, WI) supplemented with fruits, vegetables, and other edible items (e.g., nuts, cereals, seed mixtures, yogurt, peanut butter, popcorn, marshmallows, etc.) to provide variety to the diet. To further promote psychological wellbeing, animals were provided with food enrichment, structural enrichment, and manipulanda. Per WNPRC standard operating procedure (SOP), all animals received environmental enhancements including constant visual, auditory, and olfactory contact with conspecifics, the provision of feeding devices that inspire foraging behavior, the provision and rotation of novel manipulanda (e.g., Kong toys, nylabones, etc.), and enclosure furniture (i.e., perches, shelves). Environmental enhancement objects were selected to minimize chances of pathogen transmission from one animal to another and from animals to care staff. While on study, all animals were evaluated by trained animal care staff at least twice daily for signs of pain, distress, and illness by observing appetite, stool quality, activity level, and physical condition. Animals exhibiting abnormal presentation for any of these clinical parameters were provided appropriate care by clinical veterinarians. Prior to all minor/brief experimental procedures, macaques were sedated with an intramuscular dose of ketamine (10 mg/kg) and monitored regularly until fully recovered from sedation. At the end of the study, euthanasia was performed following WNPRC SOP as determined by the attending veterinarian and consistent with the recommendations of the Panel on Euthanasia of the American Veterinary Medical Association. Following sedation with ketamine (at least 15mg/kg body weight, IM), animals were administered at least 50 mg/kg i.v. or intracardiac sodium pentobarbital, or equivalent, as determined by a veterinarian. Death was defined by stoppage of the heart, as determined by a qualified and experienced individual.

## Animal care and use

*Animal cohort 1 (MCM_2wk cohort)*: Eight male Mauritian cynomolgus macaques (MCMs) purchased from Bioculture Ltd. were housed and cared for at the WNPRC. Each animal had at least one copy of the M3 MHC haplotype, and none had the M1 MHC haplotype associated with viral control (Table 1) [8,30]. All eight MCMs were infected intravenously (i.v.) with 10,000 infectious units (IUs) of barcoded SIVmac239M [70] and began receiving a daily antiretroviral therapy (ART) regimen consisting of 2.5mg/kg dolutegravir (DTG, ViiV Healthcare, Research Triangle, NC), 5.1mg/kg tenofovir disoproxil fumarate (TDF, Gilead, Foster City, CA), and 40mg/kg emtricitabine (FTC, Gilead) in 15% Kleptose (Roquette America) in water subcutaneously at two wpi. Four of these animals received a therapeutic vaccination regimen during ART described previously [32]. ART was discontinued in all animals after eight months of treatment and the animals were monitored for viral rebound. Six months after ART discontinuation, all eight animals were rechallenged i.v. with 100 TCID50 of non-barcoded SIVmac239. Eight weeks after rechallenge, all eight MCMs received a single 50 mg/kg i.v. infusion of the anti-CD8α mAb MT807R1. The rhesus macaque (RM) IgG1 recombinant Anti-CD8α [MT807R1] monoclonal antibody was engineered and produced by the Nonhuman Primate Reagent Resource (NIH Nonhuman Primate Reagent Resource Cat# PR-0817, RRID: AB_2716320). All animals were necropsied six to eight weeks after CD8α depletion.

*Animal cohort 2 (MCM_8wk cohort)*: Six male MCMs purchased from Bioculture Ltd. were housed and cared for at the WNPRC. The MHC haplotypes are summarized in Table 2. These animals had previously been exposed to dengue virus (wild type DENV-1) >5 years before SIV challenge and inoculated i.v. with plasma containing simian pegivirus [71] ~5 months after SIV challenge as part of previous projects. All six MCMs were previously inoculated intrarectally with 3,000 TCID50 of SIVmac239. One animal (cy0735) was not productively infected after this inoculation and was therefore rechallenged i.v. with 500 TCID50 of SIV-mac239 five weeks after the initial intrarectal challenge. All six MCMs received an ART regimen consisting of 2.5mg/kg DTG (APIChem, Hangzhou, Zhejiang, China), 5.1mg/kg TDF (APIChem), and 50mg/kg FTC (APIChem) in 15% Kleptose (Roquette America) in water subcutaneously daily beginning eight wpi. ART was discontinued in all MCMs after approximately 20 months of treatment and the animals were monitored for viral rebound for 81 days. Animals were transferred to a different study after this time.

*Animal cohort 3*: Five Indian RMs (three female and two male) were previously housed and cared for at the WNPRC. All RMs expressed the *Mamu-A1*001* MHC class I allele and did not express the *Mamu-B*008* or *Mamu-B*017* alleles associated with SIV control [5,6,41]. All RMs were infected i.v. with 10,000 IUs of SIVmac239M as previously described [41].

## Plasma viral load analysis

Plasma was isolated from undiluted whole blood by Ficoll-based density centrifugation and cryopreserved at -80˚C. Plasma viral loads were quantified as previously described [41]. Briefly, viral RNA (vRNA) was isolated from plasma samples using the Maxwell Viral Total Nucleic Acid Purification kit (Promega, Madison WI). Next, vRNA was reverse transcribed using the TaqMan Fast Virus 1-Step qRT-PCR kit (Invitrogen) and quantified on a LightCycler 480 instrument (Roche, Indianapolis, IN). Primers and probes are described in [72].

## Cell-associated viral RNA (vRNA) analysis

Total RNA was isolated from $1x10^6$ cryopreserved lymph node (LN) mononuclear cells (LNMC) isolated from LN biopsies using the SimplyRNA Tissue kit for the Maxwell RSC instrument (Promega, Madison, WI). RNA was then tested for both SIV *gag* and a cellular gene, *β-actin*, by RT-qPCR. Both assays were performed using the Taqman Fast Virus 1-Step Master Mix RT-qPCR kit (Applied Biosystems Inc., Carlsbad, CA) on an LC96 instrument (Roche, Indianapolis, IN) with identical experimental conditions; 600 nM each primer, 100 nM probe and 150 ng random primers (Promega, Madison, WI). Reactions cycled with the following conditions: 50˚C for 5 minutes, 95˚C for 20 seconds followed by 50 cycles of 95˚C for 15 seconds and 60˚C for 1 min. SIV: forward primer: 5′- GTCTGCGTCATPTGGTGCATT C-3′, reverse primer: 5′-CACTAGKTGTCTCTGCACTATPTGTTTTG-3′, and probe: 5′-6-carboxyfluorescein-CTTCPTCAGTKTGTTTCACTTTCTCTTCTGCG-BHQ1-3′. β-actin: forward primer: 5′-GGCTACAGCTTCACCACCAC-3′, reverse primer: 5′-CATC TCCTGCTCGAAGTCTA-3′, and probe: 5′-6-carboxyfluorescein-GTAGCAC AGCTTCTCCTTAATGTCACGC-BHQ1-3′. RNA was quantified for each reaction by interpolation onto a standard curve made up of serial tenfold dilutions of *in vitro* transcribed RNA.

Because a source of latently infected cells was not readily available, the lower limit of detection was estimated using CD8- macaque PBMCs that were infected with SIVmac239 *in vitro* at a high multiplicity of infection (MOI). CD8- cells were isolated using a nonhuman primate CD8+ T cell separation kit (Miltenyi Biotec) to collect the unlabeled cell fraction. These cells were confirmed to be SIV+ by qPCR measuring both *gag* and *ccr5* DNA. A dilution series of

the SIV+ cells was prepared in $1x10^6$ naïve macaque PBMCs. In these experiments, we can detect as few as 10 infected cells per million PBMCs.

## Quantitative Viral Outgrowth Assays (QVOAs)

QVOAs were modified from previously described work quantifying the HIV latent reservoir [73]. Briefly, peripheral blood mononuclear cells (PBMCs) were isolated from undiluted whole blood by Ficoll-based density centrifugation. Monocytes were removed by adherence via a three-hour incubation at 37°C. CD4+ T cells were then enriched using a nonhuman primate CD4+ T cell isolation kit (Miltenyi Biotec). $1x10^4$, $1x10^5$, $3x10^5$, $1x10^6$, and/or $2x10^6$ enriched CD4+ T cells were plated in 1–6 replicates per concentration per animal (depending on the available number of cells) at a 1:2 ratio with irradiated (100Gy) CEMx174 cells in R10 (RPMI 1640 medium supplemented with 10% FBS, 1% antibiotic-antimycotic [Thermo Fisher Scientific, Waltham, MA] and 1% L-glutamine [Thermo Fisher Scientific]) in the presence of phorbol myristate acetate (PMA) (0.1μg/mL, Sigma Aldrich, St. Louis, MO) and ionomycin (1μg/mL, Sigma Aldrich) and incubated overnight at 37°C. The next day, mitogens were removed by washing the contents of each well with R10, and the pellets were resuspended in R15-50 (RPMI 1640 medium supplemented with 15% fetal calf serum, 1% antibiotic/antimycotic 1% L-glutamine, and 50 U/mL of interleukin-2). Fresh CEMx174 target cells were added at a 1:1 ratio with the original plated number of enriched CD4+ T cells and plates were incubated for 2 weeks at 37°C. After two weeks, supernatant was collected and supernatant viral loads were quantified identically to plasma viral loads, described above. The calculator developed by the Siliciano lab at Johns Hopkins University was used to estimate infectious units per million cells (IUPM) for each animal by combining the number of cells plated, number of replicates at each concentration, and number of positive outcomes (positive supernatant viral loads) [74].

## Mass spectrometry quantification of Plasma ART Concentrations

ART quantification in plasma from SIV+, ART-suppressed MCMs was performed at the Antiviral Pharmacology Laboratory, University of Nebraska Medical Center, applying liquid chromatography tandem mass spectrometry (LCMS) according to previously validated methods [75]. Briefly, DTG concentrations were quantified via a methanolic precipitation extraction with stable-labeled internal standard over the range of 20–10,000 ng/mL. DTG sample extracts were detected via LCMS. TFV/FTC concentrations were quantified via an acidic protein precipitation extraction with stable-labeled internal standards over the range of 10–1,500 ng/mL. TFV/FTC sample extracts were also detected via LCMS.

## DNA extraction

Using the QIAamp DNA Mini Kit (Qiagen), DNA was extracted from previously cryopreserved PBMCs according to the manufacturer's instructions, except we eluted with 150 μL nuclease-free water. Sample DNA concentrations were measured on a Nanodrop (Thermo Fisher Scientific, MA, USA).

## Intact proviral DNA assay

To quantify frequencies of intact and defective proviral DNA, we used a previously described SIV IPDA [40,64,76] with minor modifications [77]. A step-by-step protocol and detailed explanations of our methods are in preparation (Matschke 2023, in prep). To summarize, the SIV IPDA discriminates between intact and defective proviral species by targeting three proviral amplicons across two parallel multiplexed ddPCR reactions, and intact proviruses are

defined by non-hypermutated 5' *pol* and 3' *env* sequences in the absence of 2-LTR junctions. To normalize proviral frequencies to cell equivalents and correct for DNA shearing, a third parallel ddPCR reaction targets two distinct amplicons within the host ribonuclease P p30 (*rpp30*) gene.

## IFN-γ ELISPOT assays

IFN-γ ELISPOT assays were performed using fresh PBMCs isolated from EDTA-anticoagulated blood by Ficoll-based density centrifugation, as previously described [32]. Peptides (Gag$_{386-394}$GW9, Nef$_{103-111}$RM9, and a Gag peptide pool comprising 15-mer peptides spanning the SIVmac239 Gag proteome, each overlapping by 11 amino acids [NIH HIV Reagent Program, managed by ATCC]) were selected from epitopes restricted by the *Mafa A1*063* MHC class I allele expressed on the M3 MHC haplotype [78]. Assays were performed according to the manufacturer's protocol, and wells were imaged and quantified with an ELISPOT plate reader (AID Autoimmun Diagnostika GmbH). As described previously [79], a one-tailed t-test with an α level of 0.05, where the null hypothesis was that the background level would be greater than or equal to the treatment level, was used to determine positive responses. Statistically positive responses were considered valid only if both duplicate wells contained 50 or more spot-forming cells (SFCs) per $10^6$ PBMCs. If statistically positive and $\geq$50 SFCs per $10^6$ PBMCs, reported values display the average of the two test wells with the average of all four negative control wells subtracted.

## Tetramerization of Gag$_{386-394}$GW9 monomers

Biotinylated monomers were produced by the NIH Tetramer Core Facility at Emory University (Atlanta, GA) using *Mafa*-A1*063 Gag$_{386-394}$GW9 peptides purchased from Genscript (Piscataway, NJ). *Mafa*-A1*063 Gag$_{386-394}$GW9 biotinylated monomers were tetramerized with streptavidin-PE (0.5mg/mL, BD biosciences) at a 4:1 molar ratio of monomer:streptavidin in the presence of a 1x protease inhibitor cocktail solution (Calbiochem, Millipore Sigma). Briefly, 1/5$^{th}$ volumes of streptavidin-PE were added to the monomer for 20 minutes rotating in the dark at 4˚C, and this process was repeated five times.

## Phenotype staining of T cells by flow cytometry

Previously frozen PBMCs isolated from whole blood were used to assess the quantity and phenotype of T cell populations longitudinally. Where tetramer staining was included, cells were thawed, washed with R10, and rested for 30 minutes at room temperature in a buffer consisting of 2% FBS in 1X PBS (2% FACS buffer) with 50nM dasatinib (Thermo Fisher Scientific). Cells were washed with 2% FACS buffer with 50nM dasatinib and incubated with the Gag$_{386-394}$GW9 tetramer for 45 minutes at room temperature. Cells were washed with 2% FACS buffer with 50nM dasatinib and incubated with the remaining surface markers (Table 3) for 20 minutes at room temperature. Where tetramer staining was not included, cells were thawed, washed with R10, and incubated for 20 minutes at room temperature with the surface markers indicated in Table 4 and 5. Cells were then washed with 2% FACS buffer and fixed. Where no intracellular staining was performed, cells were fixed for a minimum of 20 minutes with 2% paraformaldehyde and acquired immediately using a FACS Symphony A3 (BD Biosciences). Where intracellular staining was performed, cells were fixed using fixation/permeabilization solution (Cytofix/Cytoperm fixation and permeabilization kit, BD Biosciences) for 20 minutes at 4˚C. Cells were washed with cold 1x Perm/Wash buffer (Cytofix/Cytoperm fixation and permeabilization kit, BD Biosciences) and incubated in 1x Perm/Wash buffer containing the CTLA4 antibody (Table 3) for 20 minutes at 4˚C. Cells were washed with 1x Perm/Wash

**Table 3. Antibodies used for T cell phenotyping.**

| Antibody | Clone | Fluorochrome | Surface or intracellular |
|---|---|---|---|
| Live/Dead | | Near-infrared | Surface |
| CD3 | SP34-2 | BUV563 | Surface |
| CD4 | SK3 | BUV737 | Surface |
| CD8 | RPA-T8 | BUV395 | Surface |
| CD28 | CD28.2 | BUV661 | Surface |
| CD95 | DX2 | BV711 | Surface |
| CCR7 | 150503 | FITC | Surface |
| PD1 | EH12.2H7 | BV421 | Surface |
| LAG3 | 2561B | AF647 | Surface |
| CTLA4 | 14D3 | PE Cy7 | Intracellular |
| TIGIT | MBSA43 | PerCP-eFuor 710 | Surface |
| Gag GW9 tetramer | — | PE | Surface |

buffer and acquired immediately using a FACS Symphony A3 (BD Biosciences). The data were analyzed using FlowJo software for Macintosh (BD Biosciences, version 10.8.0). After excluding doublets and dead cells, lymphocyte populations were defined as follows: CD4+ T cells, CD3+ CD4+ CD8$\alpha$-; CD8+ T cells, CD3+ CD4- CD8$\alpha$+; Gag-specific CD8+ T cells, CD3+ CD4- CD8$\alpha$+ Gag$_{386-394}$GW9 tetramer+; CD4+CD8+ T cells, CD3+ CD4+ CD8$\alpha$+; CD4+ T$_{CM}$, CD3+ CD4+ CD8$\alpha$- CD95+ CD28+ CCR7+; CD4+ T$_{TM}$, CD3+ CD4+ CD8$\alpha$- CD95+ CD28+ CCR7-; CD4+ T$_{EM}$, CD3+ CD4+ CD8$\alpha$- CD95+ CD28- CCR7-; CD8+ T$_{CM}$, CD3+ CD4- CD8$\alpha$+ CD95+ CD28+ CCR7+; CD8+ T$_{TM}$, CD3+ CD4- CD8$\alpha$+ CD95+ CD28 + CCR7-; CD8+ T$_{EM}$, CD3+ CD4- CD8$\alpha$+ CD95+ CD28- CCR7- (S1 Fig). Cell subpopulations were excluded from analysis when the parent population contained <50 events. Of note, while the CD8$\alpha$ depleting antibody MT807R1 has been shown to mask epitopes, precluding the concomitant use of many anti-CD8$\alpha$ antibodies for immunophenotyping, the anti-CD8$\alpha$ monoclonal antibody (DK25 clone) can still discriminate CD8$\alpha$+ cells in the presence of MT807R1 [80,81]. We used the DK25 clone anti-CD8$\alpha$ monoclonal antibody to identify CD8 + T cells and NK cells after *in vivo* CD8$\alpha$ depletion.

## Intracellular cytokine staining (ICS) assays

ICS assays were performed to characterize T cell polyfunctionality similarly to published work [42,82]. Previously frozen PBMCs isolated from whole blood were thawed, washed with R10, and rested for ~6 hours in R15 (RPMI 1640 medium supplemented with 15% FBS, 1%

**Table 4. Antibodies used for T cell and NK cell phenotyping.**

| Antibody | Clone | Fluorochrome | Surface or intracellular |
|---|---|---|---|
| Live/Dead | | Near-infrared | Surface |
| CD3 | SP34-2 | AF-700 | Surface |
| CD4 | SK3 | BUV737 | Surface |
| CD8 | DK25 | PE | Surface |
| CD14 | M5E2 | APC-H7 | Surface |
| CD16 | 3G8 | BV786 | Surface |
| CD20 | 2H7 | APC-H7 | Surface |
| CD56 | B159 | FITC | Surface |
| NKG2A/C | Z199 | PE-Cy7 | Surface |

**Table 5. Antibodies used for CD4+ T cell CCR5 phenotyping.**

| Antibody | Clone | Fluorochrome | Surface or intracellular |
|---|---|---|---|
| Live/Dead | | Near-infrared | Surface |
| CD3 | SP34-2 | AF700 | Surface |
| CD4 | SK3 | BUV737 | Surface |
| CD8 | RPA-T8 | BUV395 | Surface |
| CD28 | CD28.2 | PE | Surface |
| CD95 | DX2 | PE Cy5 | Surface |
| CCR7 | 150503 | FITC | Surface |
| CCR5 | J418F1 | BV421 | Surface |

antibiotic-antimycotic [Thermo Fisher Scientific], and 1% L-glutamine [Thermo Fisher Scientific]) at 37°C in 5% $CO_2$. After resting, cells were incubated for ~16 hours at 37°C in 5% $CO_2$ with R15 alone as a negative control, or with the Gag peptide pool described above, at a final concentration of 62.5 μg/mL (0.5 μg/mL of each peptide). Two wells stimulated with 20ng/mL PMA and 1 μg/mL ionomycin were included in each batch of staining as a positive control. 5 μg/mL anti-CD28 and 5 μg/mL anti-CD49d were added during stimulation. After 90–120 minutes, 1 μg/mL Brefeldin A, 2 μM Monensin, and anti-CD107a (Table 6) were added to all cells for the remainder of the stimulation. Following the stimulation, cells were washed with 2% FACS buffer and incubated with the remaining surface markers (Table 6) for 20 minutes at room temperature. Intracellular staining, sample acquisition, and data analysis were performed as described above. All reported values are Gag-specific responses with media controls background subtracted. When Gag-specific responses were not higher than the background, values are reported as zero.

## Enzyme-Linked Immunosorbent Assays (ELISAs)

Uncoated 2HB 96-well plates (Thermo Fisher Scientific) were coated with 0.5 μg/mL SIV-mac239 gp120 (Immune Technology, New York, NY) in coating buffer (2.25mg/mL $Na_2CO_3$ and 4.395mg/mL $NaHCO_3$ in distilled $H_2O$). Plates were incubated overnight at 4°C. The next day, plates were washed with wash buffer (1X PBS with 0.05% V/V TWEEN-20 [Thermo Fisher Scientific]) and blocked in a buffer containing 1X PBS supplemented with 10% FBS for one hour at room temperature. After blocking, plates were washed and serial dilutions of plasma were added for 90 minutes at room temperature. Plasma samples were heat-inactivated at 56°C for 30 minutes, prior to use. Serial dilutions of anti-SIV gp120 monoclonal antibody (B404, NIH HIV Reagent Program, Division of AIDS, NIAID, NIH) were plated as a positive

**Table 6. Antibodies used for ICS assay.**

| Antibody | Clone | Fluorochrome | Surface or intracellular |
|---|---|---|---|
| Live/Dead | | Near-infrared | Surface |
| CD3 | SP34-2 | BUV563 | Surface |
| CD4 | SK3 | BUV737 | Surface |
| CD8 | RPA-T8 | BUV395 | Surface |
| CD107a | H4A3 | PE | Surface |
| IFN-γ | 4S.B3 | FITC | Intracellular |
| TNF-α | MAB11 | AF700 | Intracellular |
| IL-2 | MQ1-17H12 | BV605 | Intracellular |
| MIP1β | D21-1351 (RUO) | BV421 | Intracellular |

control. Next, plates were washed with wash buffer and anti-monkey IgG HRP (SouthernBiotech, Birmingham AL) was diluted 1:10,000 in blocking buffer and added to each well. Plates were incubated for one hour at room temperature. Plates were then washed with wash buffer and TMB substrate (3,3'5,5'–tetramethylbenzidine, Thermo Fisher Scientific) was added to each well. After a 15 minute incubation, HCl (1N) was added to each well to stop the reaction. ELISA plates were immediately read using a GloMax-Multi Detection System microplate reader (Promega, Madison WI) at 450 nm absorbance.

### RNA sequencing (RNAseq)

Previously cryopreserved PBMCs were thawed and CD8+ T cells were enriched by negative selection using a nonhuman primate CD8+ T cell isolation kit (Miltenyi Biotec). RNA was isolated from the enriched CD8+ T cells by TRIzol/phenol chloroform extraction with DNase treatment. RNAseq was performed by the University of Wisconsin Biotechnology Center Gene Expression Center. Briefly, the library was prepared by synthesizing cDNA via reverse transcription and amplifying cDNA by long-distance PCR using the Takara SMARTer v4 kit (Takara Bio USA, San Jose, CA). 2x150bp sequencing was performed on an Illumina NovaSeq6000 instrument. Mean +/- SD sequencing depth was 265.75M +/- 45.59M total reads. RNA sequencing data were deposited in the NCBI's Gene Expression Omnibus and are publicly available under GEO accession GSE225770.

### RNAseq analyses

BCL to FASTQ conversion was performed, then FASTQ files were used as input into the NextFlow (v22.04.5 [83]) pipeline nf-core/rnaseq pipeline (v3.8.1 [84,85]) using *Macaca fascicularis* genome version 6.0.107 (MacFas6) for alignment and annotation. Mean +/- SD alignment rate was 82.4% +/- 1.0%. Subsequent analysis was performed on the Salmon gene count matrix, with removal of genes where the total count across samples was less than 10, as well as redundant gene_name rows (Y_RNA) and all poorly annotated ENSMFAG# format genes. EBSeq [86,87] was used to determine statistical significance, due to better model fit than DESeq2, and was run for 2 condition comparison on median normalized counts data for a total of 5 iterations. Genes with a false discovery rate < 0.05 were considered significant.

### Statistical analyses

For statistical analyses in which animal groups of different species were being compared or different animal groups within a species were being compared to each other at the same time point, Mann-Whitney U tests were performed. Wilcoxon signed-rank tests were used to compare two time points from the same animal group. Chi-square tests were performed to compare frequency distributions of vDNA types in IPDAs. Where three animal groups were compared at one time point, mixed effects analyses were calculated with Geisser-Greenhouse correction and Šídák correction for multiple comparisons. When three animal groups were compared across two time points, two-way ANOVA tests were performed with Šídák correction for multiple comparisons. All statistical analyses except those for RNAseq were calculated in GraphPad Prism.

### Supporting information

**S1 Fig. T cell gating schematic.** Representative gating strategy used to evaluate frequency, memory phenotype, and exhaustion marker expression of bulk CD4+, CD8+, and CD4+CD8+ T cells, and antigen-specific CD8+ T cells.
(EPS)

**S2 Fig. Immunological changes before and after SIVmac239 rechallenge.** (A) The frequency of CD4+ T cells (left), CD8+ T cells (center), and CD4+CD8+ T cells (right) before, one week after, and six weeks after SIVmac239 rechallenge in the animals that had undetectable (circles) or detectable (triangles) viremia after rechallenge. Results are displayed for each animal individually with the lines at the median. (B) IFN-γ ELISPOT assays were performed before and one week after SIVmac239 rechallenge to assess responses to $Gag_{386-394}GW9$, $Nef_{103-111}RM9$, and a Gag peptide pool in the animals that had undetectable (circles) or detectable (triangles) viremia after rechallenge. Results are displayed for each animal individually with the lines at the median. Animals that previously received therapeutic interventions are displayed as open symbols. * $P = 0.0286$. $P$ values were calculated using Mann-Whitney U tests.
(EPS)

**S3 Fig. Anti-SIVmac239 gp120 IgG antibodies from post-ART SIV PTCs, blippers, and rebounders.** Area under the curve (AUC) of plasma anti-SIVmac239 gp120 IgG antibodies during ART (left), 10 days after ART withdrawal (center), and 12 weeks after ART withdrawal (right) in the PTCs (orange), blippers (teal), and rebounders (pink). $P$ values were calculated with a mixed-effects analysis using Geisser-Greenhouse correction and Šídák correction for multiple comparisons.
(EPS)

**S4 Fig. ICS assay gating schematic and frequencies of polyfunctional CD4+ T cells and CD8+ T cells from post-ART SIV PTCs, blippers, and rebounders on ART and after ART withdrawal.** (A) Representative gating strategy used to distinguish CD4+ and CD8+ T cell populations following overnight incubation with media alone (no stimulation) or a Gag peptide pool. (B) Representative CD107a+, IFN-γ+, and TNF-α+ subpopulations with each respective stimulus are shown for CD8+ T cells (left) and CD4+ T cells (right). (C-D) The frequency of CD8+ T cells (left) and CD4+ T cells (right) from the post-ART SIV PTCs (orange), blippers (teal), and rebounders (pink) during ART (C) and 12 weeks post ART withdrawal (D) expressing CD107a, TNF-α, and/or IFN-γ in response to *in vitro* Gag stimulation. Responses were calculated with Boolean gates identifying populations expressing permutations of CD107a, TNF-α, and IFN-γ after background subtraction.
(EPS)

**S5 Fig. Differential gene expression of CD8+ T cells from PTCs and rebounders on ART and after ART withdrawal.** (A) Heat map showing the mean-centered value of statistically significant differentially expressed genes between four PTCs (gray) and four rebounders (pink) on ART (time point 1, dark gray) and after ART withdrawal (time point 2, white). (B) Box plots showing the values for differentially expressed genes in (A) that are related to immune function. Genes were considered significant if they had a false discovery rate of < 0.05.
(EPS)

**S6 Fig. The frequency of CD8+ T cell subsets from post-ART SIV PTCs, blippers, and rebounders expressing CTLA4 on ART and after ART withdrawal.** Frequency of CTLA4+ (A) bulk, (B) CD8+ $T_{CM}$, (C) CD8+ $T_{TM}$, and (D) CD8+ $T_{EM}$ cells from the post-ART SIV PTCs (orange), blippers (teal), and rebounders (pink) during ART treatment (on ART) and 12 weeks post ART withdrawal (post ART). * $P \leq 0.05$, ** $P \leq 0.01$. $P$ values were calculated using two-way ANOVA tests with Šídák correction for multiple comparisons.
(EPS)

**S7 Fig. Frequency of Gag GW9-specific CD8+ T cells from post-ART SIV PTCs, blippers, and rebounders expressing exhaustion markers on ART and after ART withdrawal.** (A) Frequency of Gag GW9-specific CD8+ T cells, and (B-E) frequency of Gag GW9-specific CD8 + T cells expressing LAG3 (B), CTLA4 (C), PD1 (D), and TIGIT (E) in the PTCs (orange), blippers (teal) and rebounders (pink) during ART treatment (on ART) and 12 weeks post ART withdrawal (post ART). Results are displayed for each animal individually with the lines at the median. * $P \leq 0.05$, ** $P \leq 0.01$. $P$ values were calculated using two-way ANOVA tests with Šídák correction for multiple comparisons.
(EPS)

**S8 Fig. Frequency of bulk, TCM, TTM, and TEM CD4+ and CD8+ T cells expressing PD1.** (A-D) The frequency of CD4+ (left) or CD8+ (right) bulk cells (A), $T_{CM}$ (B), $T_{TM}$ (C), and $T_{EM}$ (D) expressing PD1 in the PTCs (orange), blippers (teal), and rebounders (pink) during ART treatment (on ART) and 12 weeks post ART withdrawal (post ART). * $P \leq 0.05$. $P$ values were calculated using two-way ANOVA tests with Šídák correction for multiple comparisons.
(EPS)

## Acknowledgments

Antiviral drugs were generously provided by Gilead (TDF and FTC) and ViiV Healthcare (DTG). The following reagent was obtained through the NIH HIV Reagent Program, Division of AIDS, NIAID, NIH: Peptide Pool, Simian Immunodeficiency Virus (SIV)mac239 Gag Protein, ARP-12364, contributed by DAIDS/NIAID. The Anti-CD8α [MT807R1] antibody used in this study was provided by the NIH Nonhuman Primate Reagent Resource (P40 OD028116). For positive controls in IPDAs, the following reagent was obtained through the NIH HIV Reagent Program, Division of AIDS, NIAID, NIH: Simian Immunodeficiency Virus, mac316 infected CEMx174 Cells, Clone 3D8, ARP-13239, contributed by Dr. Mario Roederer and Dr. Joseph Mattapallil. We thank the NIH Tetramer Core Facility (contract number 75N93020D00005) for generating the Mafa-A1*063 Gag$_{386-394}$GW9 biotinylated monomers. The following reagent was obtained through the NIH HIV Reagent Program, Division of AIDS, NIAID, NIH: Anti-Simian Immunodeficiency Virus (SIV) gp120 Monoclonal Antibody (B404), ARP-12146, contributed by Dr. Takeo Kuwata. The author(s) utilized the University of Wisconsin–Madison Biotechnology Gene Expression Center (Research Resource Identifier—RRID:SCR_017757) for RNA library preparation and the DNA Sequencing Facility (RRID:SCR_017759) for sequencing. ImmunityBio kindly provided the N-803 that the four vaccinated animals previously received. We are grateful to the WNPRC staff for the exceptional veterinary care provided to the animals throughout this study.

## Author Contributions

**Conceptualization:** Olivia E. Harwood, Amy L. Ellis-Connell, David H. O'Connor, Shelby L. O'Connor.

**Data curation:** Ryan V. Moriarty, Jessica D. Lang.

**Formal analysis:** Olivia E. Harwood, Lea M. Matschke, Jessica D. Lang.

**Funding acquisition:** Shelby L. O'Connor.

**Investigation:** Olivia E. Harwood, Lea M. Matschke, Alexis J. Balgeman, Abigail J. Weaver, Andrea M. Weiler, Lee C. Winchester.

**Methodology:** Lea M. Matschke.

**Resources:** Brandon F. Keele, David H. O'Connor.

**Supervision:** Courtney V. Fletcher, Thomas C. Friedrich, Matthew R. Reynolds, Shelby L. O'Connor.

**Writing – original draft:** Olivia E. Harwood.

**Writing – review & editing:** Olivia E. Harwood, Amy L. Ellis-Connell, Shelby L. O'Connor.

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
