## [Decision Letter · Decision Letter 0]

5 Jun 2023

Dear Dr. O’Connor,

Thank you very much for submitting your manuscript "CD8+ cells and small viral reservoirs facilitate post-ART control of SIV replication in M3+ Mauritian cynomolgus macaques initiated on ART two weeks post-infection" for consideration at PLOS Pathogens. As with all papers reviewed by the journal, your manuscript was reviewed by members of the editorial board and by several independent reviewers. In light of the reviews (below this email), we would like to invite the resubmission of a significantly-revised version that takes into account the reviewers' comments.

We cannot make any decision about publication until we have seen the revised manuscript and your response to the reviewers' comments. Your revised manuscript is also likely to be sent to reviewers for further evaluation.

Sincerely,

Guido Silvestri

Academic Editor

PLOS Pathogens

Susan Ross

Section Editor

PLOS Pathogens

Kasturi Haldar

Editor-in-Chief

PLOS Pathogens

orcid.org/0000-0001-5065-158X

Michael Malim

Editor-in-Chief

PLOS Pathogens

orcid.org/0000-0002-7699-2064

Reviewer's Responses to Questions

**Part I - Summary**

Reviewer #1: This manuscript by Harwood et. al describes a novel model of post treatment control (PTC) in SIV infected Mauritian cynomologous macaques (MCMs). Mechanisms underlying post treatment control have been difficult to investigate, due to its rare occurrence in both people living with HIV and in the most commonly used nonhuman primate model, SIV infection of rhesus macaques (RhMs). In MCMs that were infected with SIVmac239M and started on cART at two weeks post infection, 7 of 8 animals qualified as PTCs after cART was interrupted after 8 months. In MCMs that were infected with SIVmac239M and started cART at eight weeks post infection 2 of 6 animals were classified as PTCs upon cART release after 20 months of treatment. The authors recategorized animals from both cohorts as PTCs, blippers, or rebounders for subsequent analyses. They found that PTC status was associated with preserved CD4+ frequencies and lower frequencies of CD8+ T cells expressing exhaustion markers, but not bulk CD8 T cell frequencies, CD4+ T cell frequencies prior to cART withdrawal, frequencies of CD4+ or CD8+ T cells expressing cytokines and cytolytic markers, differential gene expression in CD8+ T cells, or cell associated viral RNA in lymph node samples on cART or post cART, or anti-SIVmac239 Env antibodies. Monoclonal antibody mediated depletion of CD8a cells in all eight of the 2 week cART group resulted in a significant increase in plasma viremia, indicating that virus suppression was mediated by CD8a cells. Interestingly, the authors showed that MCMs had significantly less total and intact vDNA at 2 weeks post infection than rhesus macaques infected with the same virus, dose, and route, which coincided with an approximate one log difference in plasma viral loads at 2wpi despite having similar viral loads at peak just a few days prior.

Given the results of this study and the limited MHC diversity in MCMs, this model provides an exciting opportunity for further investigation into the development of post treatment control. Additionally, the observation of an approximate one log reduction in plasma viral loads at 2wpi relative to SIVmac239 infected animals is another interesting potential area of research, as it suggests MCMs are able to exhibit some measure of control over acute phase SIVmac239 replication to a greater extent than RhMs. Overall, this is a well conducted study in an intriguing area of the field and the authors do a good job exploring how different aspects of this model could be further manipulated to elucidate the interplay between the virus and the immune system.

Reviewer #2: Harwood and colleagues carry out a detailed analysis of the phenomenon of post-treatment control (PTC) in Mauritian cynomolgus macaques (MCMs) infected with SIV. Unlike SIV-infected rhesus macaques (and HIV-infected humans), control of viremia after release of antiretroviral therapy (ART) was observed in a high frequency (7/8) of MCMs. Interestingly, none of these animals had MHC haplotypes previously associated with SIV control, although all expressed the M3 haplotype. Using an SIV-specific IPDA, the authors documented significantly lower level of the SIV reservoir in the MCMs compared with rhesus macaques. Furthermore, depletion of CD8alpha-expressing cells resulted in a significant rebound of viremia in the MCMs. Based on these observations, the authors conclude that the combination of relatively lower reservoirs and immune-mediated suppression contribute to the PTC phenotype in MCMs. These initial observations are provocative and support further study of the model. However, several additional studies will be required to better understand the mechanisms involved in PTC in this model. In addition, the extent to which results in this model may apply to HIV-infected humans remains uncertain.

Reviewer #3: This is an excellent manuscript from a very well-versed group that actually pioneered the Mauritian cynomolgus macaque model for HIV infection. Here they report that in an MHC-haplomatched cohort of MHC-M3+ SIVmac239+ Mauritian cynomolgus macaques (MCMs) which were treated with ART at two weeks postinfection, they observed very high levels of posttreatment control for up to 7 months after the cessation of therapy. This control was not due to a selection of a replication-incompetent virus, because they could document replication competent SIV in these postreatemnt controllers using quantitative viral outgrowth assays. They also report that in vivo depletion of CD8α+ cells induced rebound in all animals, and suggest that this could indicate that the observed PTC was mediated, at least in part, by CD8α+ cells. They also performed IPDA to measure the size of the intact viral reservoir and report that in these macaques there are significantly smaller viral reservoirs two wpi than a cohort of identically infected rhesus macaques, a population that rarely develops PTC. They conclude that an unusually small viral reservoir is a hallmark among SIV+ MCMs. They also established that PTC was associated with a reduced frequency of CD4+ and CD8+ lymphocyte subsets expressing exhaustion markers. Together, these results suggest a combination of small reservoirs and immune-mediated virus suppression contribute to PTC in MCMs.

The manuscript is concise, well written that results are of potentially high impact and, altogether my recommendation is of acceptance after several minor modifications.

Reviewer #4: This study by Harwood et al. reports a high frequency of post-treatment controllers in a cohort of SIV-infected MCM expressing at least one copy of the M3 MHC haplotype, treated with ART for ~8 months starting 2 weeks post-infection (wpi). Seven out of eight MCM controlled viremia for extended period of times after ATI despite the demonstrated presence of replication-competent viruses in the reservoir. Following rechallenge with an isogenic virus, some of these PTC were shown to maintain undetectable viremia while CD8+ cell depletion inevitably led to a rebound in viremia. Additional viral loads data are presented from other cohorts of RMs and MCM initiated on ART during chronic infection. Animals from both cohorts of MCM were then regrouped based on ATI outcome to interrogate the parameters associated with PTC.

The manuscript contains a lot of interesting data supporting the specificity of SIV immunopathogenesis in Mauritanian Cynomolgus macaques previously suggested by others. However, data interpretation is at time confusing and complicated by the design of some analyses and by their representation. Most notably, the validity of the comparative analyses is questionable due to important confounding factors including genetic background, route of infection and infectious dose. Considering these limitations, the authors should be more measured in their interpretation of the data and conclusions.

**Part II – Major Issues: Key Experiments Required for Acceptance**

Reviewer #1: One result that I found particularly interesting, but don’t feel the authors discussed enough, was the finding that there was no difference in lymph node cell associated viral RNA post cART across the PTC, blipper, and rebound groups, despite distinct plasma viral load profiles. This suggests that there is an effective immune response that is containing viral replication in the PTCs and blippers in this compartment that isn’t present in the rebounders. While the authors had lymph node homogenates for the vRNA measurements, they do not address whether assays for cellular responses could be conducted on additional LN cells from this biopsy. Understanding the lymph node resident CD8+ T cell response is key given this result, especially in light of the fact that analyses of blood CD8+ T cells only showed a difference in expression of exhaustion markers, but not much of a difference in expression of functional cytokines and cytolytic markers. I would encourage the authors to conduct these analyses on cryopreserved LN cell samples if they are available, and if they are not to add a paragraph in the discussion addressing the discordant pVL vs LN vRNA result and the possible role of tissue resident CD8+ T cells.

Reviewer #2: 1) Page 6, lines 114-115: The authors state for the 2-week group they only included animals carrying the M3 haplotype, but do not provide any justification for this decision. Was the inclusion of only M3 animals in the 2-week cohort by chance or by design? And if by design, what was the rationale? In several instances (p. 2, line 24; p. 16, line 356) the authors qualify their results to M3+ haplotype MCMs, but they present no data (and there is no discussion) on whether this is a phenomenon specific to M3 haplotyope MCM or whether it may occur in other MCM as well. This issue should be explicitly addressed by the authors.

2) Although the authors present data and implicate cellular immune responses in PTC, no data are presented regarding SIV-specific neutralizing antibody responses. Such data would considerably strengthen the manuscript.

3) Given the data suggesting a smaller intact reservoir in SIV-infected MCMs compared with rhesus macaques, efforts to analyze the size of SIV-susceptible target cells in both species would be of interest, including analysis in peripheral blood, lymph node, and gut-associated lymphoid tissue. Do the two species differ in percentages of memory, activated CD4+ T cells or CCR5+/CD4+ T cells?

4) Page 9, lines 194-199: Are the differences in the percentage of animals exhibiting sustained plasma viremia significantly different between the 2-week MCM cohort and the 8-week MCM cohort?

Reviewer #3: None

Reviewer #4: 1- While it is not directly shown, it seems that viremia might have peaked earlier in the MCM than in the RMs. The lower levels of total and intact DNA in MCMs at 2wpi could thus reflect their lower plasma viral load - which maybe be controlled faster due to intrinsic differences in T cells responses. It would be interesting to compare total and intact DNA levels between MCM and RM at the time of peak viremia rather than at 2wpi. Also, showing the longitudinal PVL in the group of RMs (at least 1 and 2wpi) would be informative. The comparison between groups is limited as the RMs did not receive ART.

2- The cohort of MCM initiated on ART at 8 wpi presents several differences with the group of MCM initiated on ART at 2 wpi that limit its value as a comparative group. Considering the differences in term of route of infection, possibly infectious dose, and the fact that pre-ART levels of total and intact DNA were measured at 2 different time points (2wpi during peak viremia for cohort 1 and at 8wpi for cohort 2, it seems difficult to conclude that “A longer duration of SIV infection prior to ART does not increase the frequency of cells with intact proviral DNA but increases the likelihood of viral rebound” (title of the 5th result paragraph). The same comment applies to the next paragraph title “despite similar reservoir size” and to several other sentences throughout the manuscript claiming a smaller reservoir in cohort 1.

**Part III – Minor Issues: Editorial and Data Presentation Modifications**

Reviewer #1: 1. Line 113: Suggest this sentence include a reference to M6, e.g. “None of these MCMs possessed the M1 or M6 MHC haplotypes associated with SIV control (Table 1).”

2. Line 116, lines 339-346: It would be nice to include at least a sentence or two describing the therapeutic vaccination given to four of the animals, to provide a bit of context for the reader, rather than making them access the reference. Namely, timing of vaccination relative to infection, relative to cART release, and results of vaccination. While the current results did not seem to suggest that the vaccination had an impact on PTC outcome, the reader should have enough information to think about whether this treatment may have impacted the current study in any way.

3. Line 147: Suggest adding the phrase “in all animals” after “increase in SIV viremia”.

Reviewer #2: 1) Page 6, lines 115-116: the specific therapeutic vaccination regimen should be briefly described in the text.

2) The issue of whether the CD8alpha-specific monoclonal antibody used for flow cytometry recognizes a distinct epitope from the MT807R1 antibody used for in vivo depletion should be explicitly noted.

Reviewer #3: My minor observations are:

-the CD8+ cell depletion experiments induce a massive proliferation of the CD4+ T cells that can be monitored by the measurement of the CD4+ T cell fraction expressing the proliferation marker Ki—67. This increased CD4 proliferation could contribute at least in part to virus reactivation upon depletion. In my opinion, it is important that we monitor the dynamics of the KI-67 expression by the T cells after the CD8 cell depletion.

-one of the reasons for a lower viral reservoir in the MCMs may be that the virus is less well adapted in the MCMs than in the rhesus macaques. In the past there were several instances in which, after a relatively successful acute replication, cross-species transmitted viruses were controlled by the macaque host [i.e., SIVrcm by cynomolgous macaques (Marx), SIVagm by the rhesus macaques (Pandrea) etc]. It might be great to see the dynamics of interferon production during the acute infection in these monkeys, although it is possible that samples might be limited. These issues should at least be discussed.

Anyway, the authors should be commended for the quality of their work.

Reviewer #4: 1- Regarding the role of the CD8+ cells in the post-treatment control of viremia, it would be informative to include the longitudinal levels of CD8+ T cells and NK cells represented in parallel with the viremia for each MCM individually to show if control is restored when CD8+ T cells return.

2- It would be best to keep the individual symbol representation of each MCM from cohort 1 and cohort 2 throughout all figures.

3- Figure 3A: the schematic indicates that cohort 2 was initiated on ART at 2wpi instead of 8wpi

4- In the discussion, the authors state that “the frequency of bulk and SIV-specific T cells subsets expressing ICIs declined after ART withdrawal in PTC”. This should be rephrased to accurately describe the results.

5- In the discussion, the authors state that Bender et al reported levels of intact DNA 10 times higher in RMs than in humans. This article reports levels 10 times higher after 36 weeks of ART in RMs infected with SIVmac251 but similar levels of intact DNA between HIV-infected individuals on long term ART and RM infected with SIVmac239 on ART.

PLOS authors have the option to publish the peer review history of their article (what does this mean?). If published, this will include your full peer review and any attached files.

Reviewer #1: No

Reviewer #2: No

Reviewer #3: **Yes: **Cristian Apetrei

Reviewer #4: No
---

## [Decision Letter · Decision Letter 1]

12 Sep 2023

Dear Dr. O’Connor,

We are pleased to inform you that your manuscript 'CD8+ cells and small viral reservoirs facilitate post-ART control of SIV replication in M3+ Mauritian cynomolgus macaques initiated on ART two weeks post-infection' has been provisionally accepted for publication in PLOS Pathogens.

Best regards,

Guido Silvestri

Academic Editor

PLOS Pathogens

Susan Ross

Section Editor

PLOS Pathogens

Kasturi Haldar

Editor-in-Chief

PLOS Pathogens

orcid.org/0000-0001-5065-158X

Michael Malim

Editor-in-Chief

PLOS Pathogens

orcid.org/0000-0002-7699-2064

Reviewer Comments (if any, and for reference):

Reviewer's Responses to Questions

**Part I - Summary**

Reviewer #2: The authors' detailed revisions have appropriately addressed my previous concerns.

Reviewer #3: The authors addressed my comments in the revised version

Reviewer #4: All comments have been addressed and the manuscript was largely improved from its original version. This reviewer particularly appreciate the addition of new graphs to Figure 2.

**Part II – Major Issues: Key Experiments Required for Acceptance**

Reviewer #2: (No Response)

Reviewer #3: None

Reviewer #4: (No Response)

**Part III – Minor Issues: Editorial and Data Presentation Modifications**

Reviewer #2: (No Response)

Reviewer #3: None

Reviewer #4: (No Response)

PLOS authors have the option to publish the peer review history of their article (what does this mean?). If published, this will include your full peer review and any attached files.

Reviewer #2: No

Reviewer #3: **Yes: **Cristian Apetrei

Reviewer #4: No

---

## [Editor Report · Acceptance letter]

20 Sep 2023

Dear Dr. O’Connor,

We are delighted to inform you that your manuscript, "CD8+ cells and small viral reservoirs facilitate post-ART control of SIV replication in M3+ Mauritian cynomolgus macaques initiated on ART two weeks post-infection," has been formally accepted for publication in PLOS Pathogens.

Best regards,

Kasturi Haldar

Editor-in-Chief

PLOS Pathogens

orcid.org/0000-0001-5065-158X

Michael Malim

Editor-in-Chief

PLOS Pathogens

orcid.org/0000-0002-7699-2064